# A new patient-derived iPSC model for dystroglycanopathies validates a compound that increases glycosylation of α-dystroglycan

Jihee Kim[1,2,†], Beatrice Lana[1,2,†], Silvia Torelli[3], David Ryan[4], Francesco Catapano[3], Pierpaolo Ala[3], Christin Luft[5], Elizabeth Stevens[3], Evangelos Konstantinidis[1,2], Sandra Louzada[4], Beiyuan Fu[4], Amaia Paredes-Redondo[1,2], AW Edith Chan[6], Fengtang Yang[4], Derek L Stemple[4] [ID], Pentao Liu[4], Robin Ketteler[5] [ID], David L Selwood[6] [ID], Francesco Muntoni[3,7,‡] & Yung-Yao Lin[1,2,*,‡] [ID]

## Abstract

Dystroglycan, an extracellular matrix receptor, has essential functions in various tissues. Loss of α-dystroglycan-laminin interaction due to defective glycosylation of α-dystroglycan underlies a group of congenital muscular dystrophies often associated with brain malformations, referred to as dystroglycanopathies. The lack of isogenic human dystroglycanopathy cell models has limited our ability to test potential drugs in a human- and neural-specific context. Here, we generated induced pluripotent stem cells (iPSCs) from a severe dystroglycanopathy patient with homozygous FKRP (fukutin-related protein gene) mutation. We showed that CRISPR/Cas9-mediated gene correction of FKRP restored glycosylation of α-dystroglycan in iPSC-derived cortical neurons, whereas targeted gene mutation of FKRP in wild-type cells disrupted this glycosylation. In parallel, we screened 31,954 small molecule compounds using a mouse myoblast line for increased glycosylation of α-dystroglycan. Using human FKRP-iPSC-derived neural cells for hit validation, we demonstrated that compound 4-(4-bromophenyl)-6-ethylsulfanyl-2-oxo-3,4-dihydro-1H-pyridine-5-carbonitrile (4BPPNit) significantly augmented glycosylation of α-dystroglycan, in part through upregulation of LARGE1 glycosyltransferase gene expression. Together, isogenic human iPSC-derived cells represent a valuable platform for facilitating dystroglycanopathy drug discovery and therapeutic development.

**Keywords** CRISPR; fukutin-related protein; high-throughput screening; human-induced pluripotent stem cells; α-dystroglycan
**Subject Categories** Cell Adhesion, Polarity & Cytoskeleton; Molecular Biology of Disease; Post-translational Modifications & Proteolysis

## Introduction

Post-translational processing of dystroglycan is crucial for its function as an extracellular matrix (ECM) receptor in a variety of fetal and adult tissues. The dystroglycan precursor is cleaved into non-covalently associated α- and β-subunits, forming an integral component of a multiprotein complex. The mature α-dystroglycan is a heavily glycosylated peripheral membrane protein with molecular weight range from 100 to 156 kDa, dependent on tissue-specific glycosylation. The β-dystroglycan is a 43-kDa transmembrane protein that links to the actin cytoskeleton via interaction with dystrophin [1]. Defective O-linked glycosylation of α-dystroglycan is a common pathological hallmark associated with number of genetic syndromes, encompassing symptoms from muscular dystrophies to ocular defects, cognitive deficits, and structural cortical malformations (cobblestone lissencephaly) in the central nervous system (CNS). This group of autosomal recessive disorders are commonly referred to as secondary dystroglycanopathies [2,3]. Currently, there is no cure for dystroglycanopathies. The effort of identifying causative gene mutations in dystroglycanopathies has shed light on a novel mammalian glycosylation pathway [4,5].

To date, at least 18 genes have been implicated in dystroglycanopathies and their products elaborate sequentially the functional glycosylation of α-dystroglycan (core M3 glycan) required

1 Centre for Genomics and Child Health, Blizard Institute, Barts and the London School of Medicine and Dentistry, Queen Mary University of London, London, UK
2 Stem Cell Laboratory, National Bowel Research Centre, Blizard Institute, Barts and the London School of Medicine and Dentistry, Queen Mary University of London, London, UK
3 UCL Great Ormond Street Institute of Child Health, London, UK
4 Wellcome Sanger Institute, Hinxton, Cambridge, UK
5 MRC Laboratory for Molecular Cell Biology, University College London, London, UK
6 The Wolfson Institute for Biomedical Research, University College London, London, UK
7 NIHR Biomedical Research Centre at Great Ormond Street Hospital, London, UK
*Corresponding author. Tel: +44 2078 822339; E-mail: yy.lin@qmul.ac.uk
†These authors contributed equally to this work as first authors
‡These authors contributed equally to this work as senior authors

for binding with (ECM) ligands, e.g., laminins, perlecan, and neurexin (Fig 1A) [4,5]. The genes involved include those for dolichol-phosphate-mannose synthesis: *GMPPB, DPM1, DPM2, DPM3* and *DOLK* [6–13]; genes required for *O*-mannosylation and subsequent sugar addition: *POMT1, POMT2, POMGNT1, POMGNT2/GTDC2,* and *B3GALNT2* [14–21] and an *O*-mannose-specific kinase gene: *POMK/SGK196* [21,22]. Recent studies have demonstrated that the *ISPD* gene encodes a CDP-ribitol pyrophosphorylase that generates the reduced nucleotide sugar for the addition of tandem ribitol-5-phosphate to α-dystroglycan by ribitol-5-phosphate transferases, encoded by the *FKTN* and *FKRP* genes (Fig 1A) [23–30]. Furthermore, *TMEM5* and *B4GAT1/B3GNT1* genes encode enzymes to prime the phospho-ribitol with xylose and then glucuronic acid [27,31–34]. *LARGE1* encodes a bifunctional enzyme to synthesize the subsequent extension of xylose-glucuronic acid disaccharide repeats that function as the binding sites for laminins (IIH6 epitope) (Fig 1A) [35,36]. Overexpression of *LARGE1* in human cell lines and transgenic mice results in enhanced functional glycosylation of α-dystroglycan and similar genetic strategies have been proposed as a potential therapeutic approach in dystroglycanopathies [37,38].

Among dystroglycanopathies, allelic *FKRP* mutations are the more prevalent and cause a wide spectrum of clinical severities that range from the mild late-onset limb-girdle muscular dystrophy without neurological deficits (e.g. LGMD2I) to congenital muscular dystrophy with severe CNS abnormalities (e.g. Walker–Warburg syndrome) [39]. Malformations of cortical development are pathological features commonly encountered in patients at the severe end of the dystroglycanopathy spectrum [40,41]. Although murine models have been widely used to study mammalian cortical malformations, successful translation from animal models to patients has been limited. To facilitate drug discovery and development for dystroglycanopathy, there is an unmet need for isogenic, physiology-relevant human cell models to test potential drug candidates in a human- and neural-specific context. Recent studies have demonstrated success in using patient-specific induced pluripotent stem cells (iPSCs) to model a variety of neurodevelopment and neurodegenerative disease and their use in high-throughput screening [42,43]. We hypothesized that patient-specific iPSC-derived neural cells can recapitulate pathological hallmarks in dystroglycanopathies and can be used for testing potential drug candidates.

Here, we generated the first human FKRP-iPSCs from a dystroglycanopathy patient with severe CNS abnormalities. We carried out CRISPR/Cas9-mediated genome editing [44,45] to correct the *FKRP* mutation in FKRP-iPSCs or knock-in the *FKRP* mutation in unrelated wild-type iPSCs. We showed that, for the first time, targeted gene correction of *FKRP* restores tissue-specific α-dystroglycan functional glycosylation in iPSC-derived neural cells, whereas targeted gene mutation of *FKRP* in wild-type cells leads to the expected disruption of functional glycosylation of α-dystroglycan. In parallel, we screened 31,954 compounds for increased glycosylation of α-dystroglycan in the H2K 2B4 mouse myoblast cell line [46]. We identified and validated a compound, 4-(4-bromophenyl)-6-ethylsulfanyl-2-oxo-3,4-dihydro-1H-pyridine-5-carbonitrile (4BPPNit; PubChem CID 2837349), that significantly augmented tissue-specific functional glycosylation of α-dystroglycan in human FKRP-iPSC-derived neural cells. This effect appears to be secondary to the upregulated *LARGE1* gene expression, suggesting a

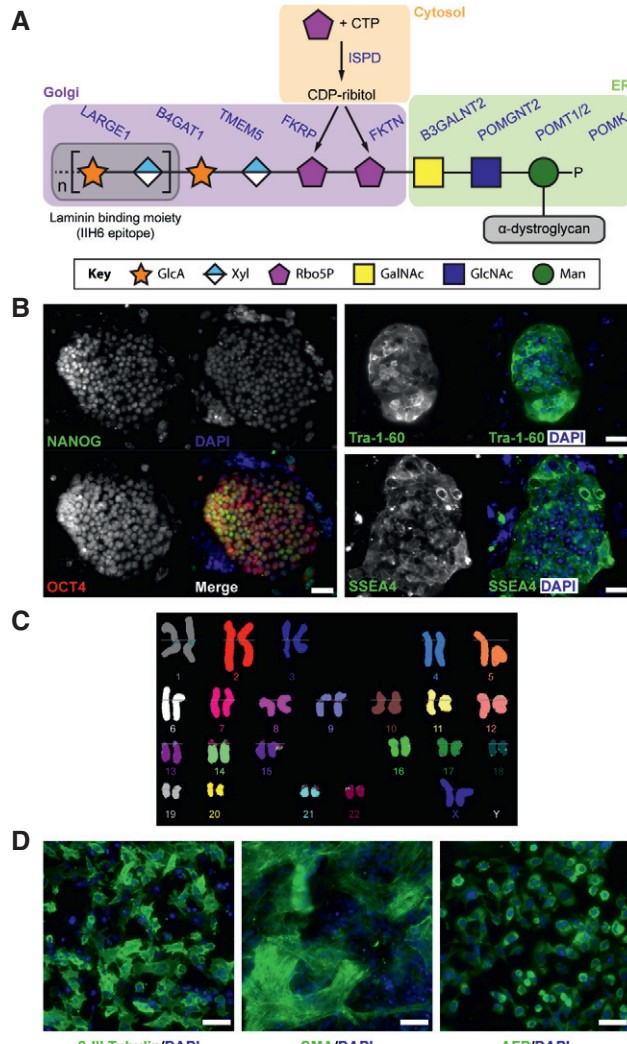

**Figure 1. Functional glycosylation of α-dystroglycan and characterization of dystroglycanopathy patient-specific iPSCs.**

A  Current model of the core M3 functional glycan structure on α-dystroglycan and enzymes involved in its synthesis. ECM ligands, such as laminins, bind to the Xyl-GlucA disaccharide repeats (IIH6 epitope). Man, mannose; GlcNAc, *N*-acetylglucosamine; GalNAc, *N*-acetylgalactosamine; Rbo5P, ribitol-5-phosphate; Xyl, xylose; GlcA, glucuronic acid.

B  Representative images of immunostaining demonstrate that FKRP-iPSCs express specific pluripotency-associated markers, including NANOG, OCT4, Tra-1-60, and SSEA4.

C  FKRP-iPSCs have a normal karyotype.

D  *In vitro* differentiation of FKRP-iPSCs to cells representing ectoderm (β-III Tubulin, Tuj1), mesoderm (SMA, smooth muscle actin), and endoderm (AFP, α-fetoprotein).

Data information: Scale bars, 50 μm.

mechanism by which 4BPPNit leads to increased functional glycosylation of α-dystroglycan. This proof-of-concept study demonstrates that isogenic, physiology-relevant human iPSC-based platform can contribute to facilitating drug development for dystroglycanopathies.

  

# Results

## Generation and characterization of FKRP-iPSCs derived from a dystroglycanopathy patient with CNS abnormalities

We obtained dermal fibroblasts from a female individual, previously diagnosed with congenital muscular dystrophy with CNS abnormalities, including marked cognitive delay, microcephaly, cerebellar cysts, and cerebellar dysplasia. Clinical features summarized in Table 1 are consistent with previously reported descriptions [40,47,48]. Genetic analysis revealed homozygous *FKRP* c.1364C>A (p.A455D) mutations in this affected individual; both parents are heterozygous carriers of this mutation. To generate dystroglycanopathy iPSCs, we reprogrammed the patient's fibroblasts carrying the *FKRP* c.1364C>A mutations using a six-factor reprogramming technology based on a doxycycline-inducible system [49]. The initial characterization of five independent FKRP-iPSC clonal lines by qPCR revealed the gene expression of pluripotency markers, such as *NANOG, OCT4* and *REX1* (Appendix Fig S1). Among these clonal iPSC lines, we focused on the FKRP-iPSC line (clone 1-6). Immunocytochemistry confirmed the expression of pluripotency markers, such as NANOG, OCT4, Tra-1-60 and SSEA4 in this FKRP-iPSC line (Fig 1B), which exhibited the normal karyotype (Fig 1C). In addition, *in vitro* differentiation of the FKRP-iPSCs formed embryoid bodies with cell types representing all three embryonic germ layers, confirmed by immunocytochemistry in specific cell lineage markers, such as β-III tubulin (Tuj1) for ectoderm, smooth muscle actin (SMA) for mesoderm, and α-fetoprotein (AFP) for endoderm (Fig 1D).

## Targeted gene correction of FKRP-iPSCs using CRISPR/Cas9-mediated genome editing

A common issue of patient-specific iPSCs in disease modeling is the lack of appropriate isogenic control cells, causing concerns about the effect of genetic backgrounds on phenotypic variability [50]. To overcome this issue, we applied a precise genome-editing strategy to correct the *FKRP* c.1364C>A (p.A455D) mutation using site-specific endonuclease CRISPR/Cas9 stimulated homologous recombination [44,45], which consists of a single-guide RNA (sgRNA), the Cas9 nuclease, and a donor-targeting vector. To minimize potential off-target effects, we used an optimized computational algorithm (http://crispr.mit.edu) to identify appropriate sgRNAs for utilizing the CRISPR/Cas9 to generate DNA double-strand breaks (DSB) near the *FKRP* mutation (Fig 2A). The sgRNA used in this study has high on-target and low off-target scores (Appendix Table S1). A transposon-based selection cassette *piggyBac* (*PGK-puroΔtk*) was used in the targeting donor vector to enable both positive and negative selections to facilitate the screening process for edited events [51].

We PCR-amplified two 1-kb fragments flanking the CRISPR/Cas9 target site close to the *FKRP* c.1364C>A allele, which was simultaneously corrected (*FKRP c.1364C*). The two fragments (homology left and right arms) flanking a *piggyBac* (*PGK-puroΔtk*) selection cassette were assembled together into a targeting donor vector (Fig 2B). DNA sequences at the junctions between *piggyBac* (*PGK-puroΔtk*) selection cassette and homology arms were modified to accommodate TTAA sequences, yet coding the same amino acids after excision of the selection cassette (Fig 2B). We electroporated the site-specific CRISPR/Cas9 plasmids with the targeting donor vector (*FKRP* c.1364C) into the iPSCs. By positive selection with puromycin, iPSCs with an integrated donor vector formed puromycin-resistant clones, which were picked for rapid PCR genotyping (Appendix Fig S2A and B). We identified 3 homozygously and 27 heterozygously targeted independent clones (Appendix Table S2), which were confirmed by sequencing (data not shown). To excise the *piggyBac* (*PGK-puroΔtk*) selection cassette, 2 homozygously targeted *FKRP* (c.1364C)-iPSC clones (5.19 and 3.16) were electroporated with *piggyBac* transposase expression plasmid, followed by negative selection in culture media containing 1-(2-Deoxy-2-fluoro-β-D-arabinofuranosyl)-5-iodo-2,4(1H,3H)-pyrimidinedione (FIAU), a thymidine analogue that is processed to toxic metabolites in the presence of *piggyBac* (*PGK-puroΔtk*). PCR genotyping identified 11 biallelicly corrected-iPSC clones that had the selection cassette completely excised without re-integration (Appendix Fig S2C and Table S2). The biallelicly corrected-iPSC clones (5D17, 5D23, and 3B17) were sequenced to confirm a precise correction of the *FKRP* mutation and the engineered selection cassette excision site (Fig 2C). The biallelicly corrected-iPSC lines retained normal karyotypes (Fig 2D). In addition, sequencing of the top 5 potential off-target sites (Appendix Table S1) confirmed that no mutations were introduced during genome editing (data not shown).

## Generation of neural stem cells and cortical neurons from FKRP- and CRISPR/Cas9 corrected-iPSCs

To investigate the potential of iPSCs for modeling neural pathogenesis in dystroglycanopathies, we used a serum-free neural induction medium to derive primitive neural stem cells (NSCs) from iPSCs [52]. Immunocytochemistry confirmed that NSCs derived from FKRP- and CRISPR/Cas9 corrected-iPSC lines expressed the classic NSC markers, including SOX1, SOX2, and nestin (Fig 3A and B). In terms of the efficiency of neural induction, no discernible difference

**Table 1.** Clinical features of the affected individual with *FKRP* mutations.

|  | Subject |
| --- | --- |
| *FKRP* variants | c.1364C>A (p.A455D) |
| Zygosity | Homozygous |
| Sex | Female |
| Age at assessment | 11 months |
| Symptoms at presentation | Hypotonia; floppiness; paucity of spontaneous movements |
| Feeding difficulties | From first few months of life |
| Respiratory difficulties | Frequent chest infection from 6 months |
| Orthopedic complications | Hip subluxation |
| Max functional achievement | Partial antigravity in arms and legs; unable to sit |
| Serum CK (NR < 200 U/l) | 5000-7090 |
| CNS function | Partially alert |
| Head circumference | Microcephaly |
| MRI | Cerebellar cysts and cerebellar dysplasia |
| Muscle biopsy | Dystrophic; severely reduced IIH6 staining |

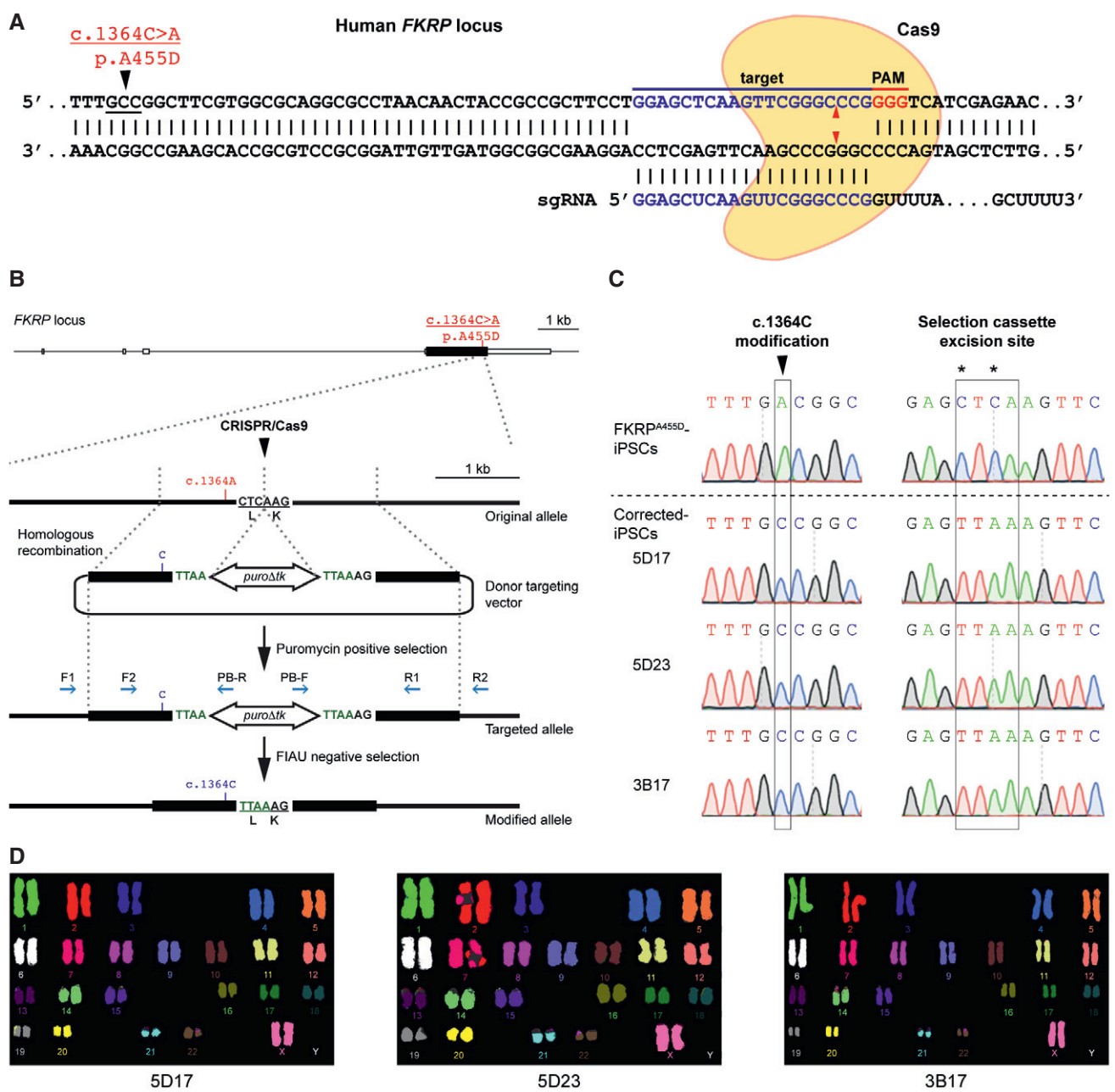

**Figure 2. Targeted gene correction of FKRP^A455D-iPSCs by CRISPR/Cas9-mediated genome editing.**

A  Cas9 protein and the specific sgRNA targeting the human *FKRP* locus. The *FKRP* c.1364C>A (p.A455D) mutation is 43 bases upstream of the sgRNA target sequences. Red arrowheads indicate putative cleavage site. PAM, protospacer-adjacent motif.

B  A schematic diagram shows the genome-editing strategy based on CRISPR/Cas9-stimulated homologous recombination, followed by positive selection with puromycin and negative selection with FIAU. Homology left and right arms on the targeting donor vector are indicated as black boxes, flanking the *piggyBac* (*PGK-puroΔtk*) selection cassette, which is under the control of *PGK* promoter. PCR genotyping primers are shown as blue arrows. Note that the TTAA sequences are designed to accommodate the selection cassette excision sites, yet code the same amino acids.

C  Sequence analysis shows precise biallelic correction of the *FKRP* mutation in three independently corrected-iPSC clones (5D17, 5D23, and 3B17), compared with their parental FKRP^A455D-iPSCs. Selection cassette excision sites are identified in the corrected-iPSC lines.

D  CRISPR/Cas9 corrected-iPSC lines show a normal karyotype.

was observed between NSCs derived from FKRP- and the three corrected-iPSC lines, 5D17, 5D23, and 3B17 (Fig 3C and D). Quantification of SOX1- and SOX2-positive NSCs showed >99% efficiency from both FKRP- and three corrected-iPSC lines (Fig 3C and D).

Subsequently, we directed the differentiation of FKRP- and corrected-NSC lines toward cortical projection neurons using a well-established protocol, which recapitulates important stages in human cortical development [53]. After switching to the Neural

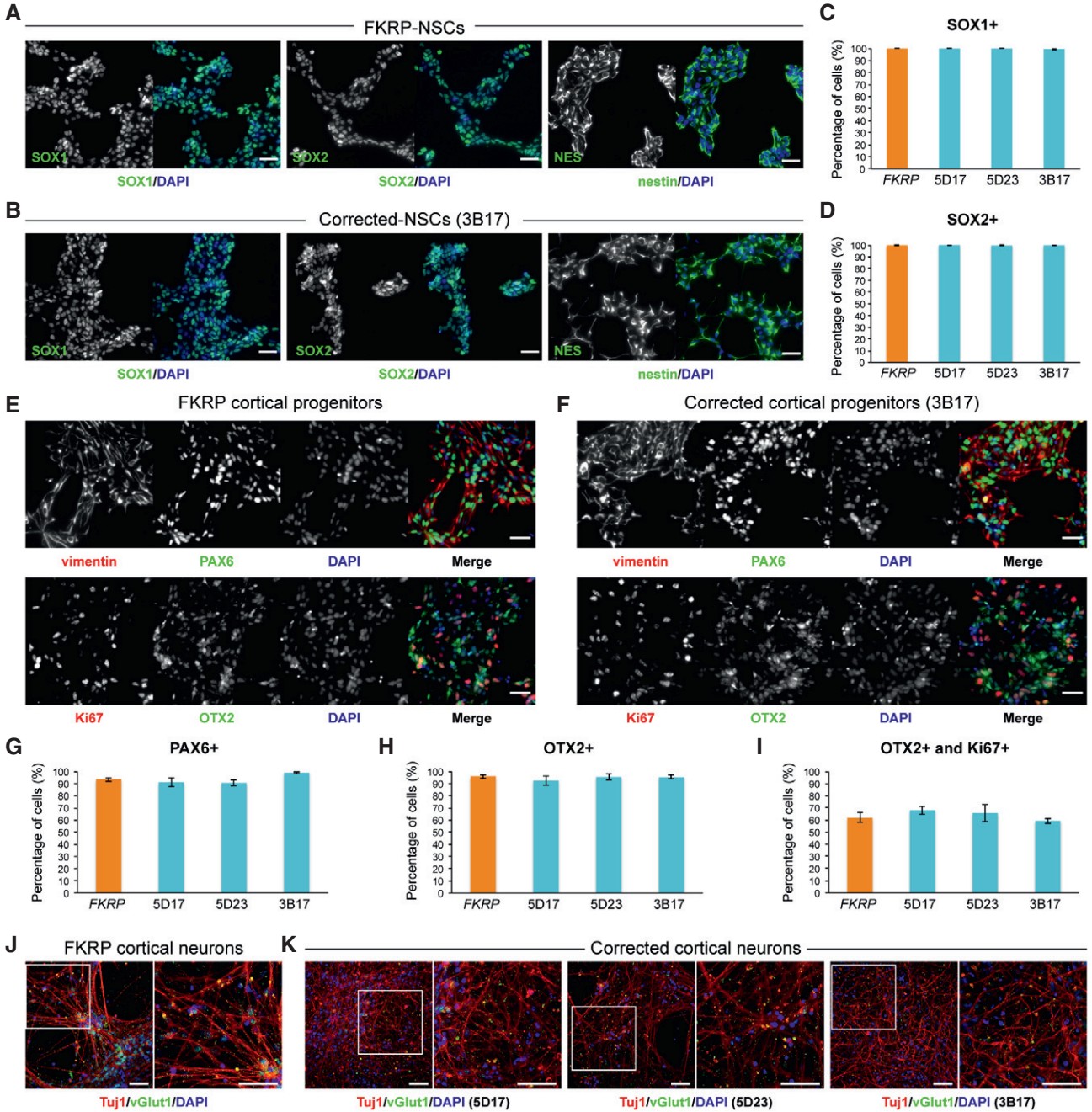

**Figure 3. Characterization of NSCs and cortical neurons derived from FKRP- and CRISPR/Cas9 corrected-iPSCs.**

A, B   Representative images of NSCs derived from FKRP- and corrected-iPSC lines expressing SOX1, SOX2, and nestin.

C, D   Quantification of percentage of SOX1$^+$ (C) and SOX2$^+$ (D) cells in culture. The efficiency of neural induction is more than 99% in FKRP- and corrected-iPSC (5D17, 5D23, and 3B17) lines. Data are mean $\pm$ s.d. $n$ = 4 technical replicates.

E, F   FKRP- and corrected-NSC lines can be further differentiated to cortical neural progenitor cells, expressing PAX6, OTX2, and vimentin.

G–I   Quantification of percentage of PAX6$^+$ (G) and OTX2$^+$ (H) cells in culture. About 91-98% of cells derived from *FKRP*, 5D17, 5D23, and 3B17 NSC lines express PAX6 (G). About 93-96% of cells derived from *FKRP*, 5D17, 5D23, and 3B17 NSC lines express OTX2 (H). Of the OTX2$^+$ population, about 60-67% cells are also Ki67$^+$ cycling progenitors (I). Data are mean $\pm$ s.d. $n$ = 4 technical replicates.

J, K   Glutamatergic projection neurons derived from FKRP and corrected (5D17, 5D23, and 3B17) progenitor cells. The vast majority of neurons contain vGlut1$^+$ punctae in their neurites (labeled by Tuj1). Right panels are enlarged images from the insets of left panels.

Data information: Scale bars, 50 μm.

Maintenance Medium for one week, we confirmed the identity of FKRP- and corrected-iPSC-derived cortical stem and progenitor cells, which expressed the classic cortical stem cell markers, PAX6, OTX2, and vimentin (Fig 3E and F); the proliferating cells were Ki67-positive (Fig 3E and F). The efficiency of cortical induction between FKRP and three lines of corrected cortical progenitors were very similar. The PAX6$^+$ cells in culture were about 91-98%, and OTX2$^+$ cells are about 93-96% (Fig 3G and H); 60-67% of OTX2$^+$ cells were also Ki67$^+$ cycling progenitors (Fig 3I). After three weeks in Neural Maintenance Medium, FKRP and corrected progenitor-derived cells showed punctate staining of vesicular glutamate transporter 1 (vGlut1) in their neurites labeled by neuron-specific tubulin, Tuj1 (Fig 3J and K). This confirmed the generation of glutamatergic projection neurons during cortical neurogenesis in culture. Although the FKRP(A455D) variant does not appear to be required during early neurogenesis, we cannot exclude that this mutation might be responsible for pathological phenotypes at a later stage.

### Tissue-specific functional glycosylation of α-dystroglycan is restored in cortical neurons derived from CRISPR/Cas9-corrected-iPSCs

Next, we investigated the presence of functional glycosylation of α-dystroglycan in cortical neurons derived from our corrected-iPSC lines using immunoblotting with the IIH6 antibody, which recognizes the laminin-binding glyco-epitope on α-dystroglycan [54]. Wild-type mouse muscle and brain lysates were used as positive controls to show differential glycosylation of α-dystroglycan in a tissue-specific manner. In addition, lysate of cortical neurons derived from a non-isogenic, human wild-type iPSC line (WT-iPSCs) was included [49]. We showed that the molecular weight of glycosylated α-dystroglycan in WT-iPSC-derived cortical neurons is similar to that in mouse brain

lysate (~120 kDa) and less than that in muscle lysate (~156 kDa) (Fig 4A), consistent with previously reported tissue-specific glycosylation of α-dystroglycan [1]. Furthermore, IIH6 reactivity on immunoblot was almost not detected in cortical neurons derived from FKRP-iPSCs (Fig 4A and B). In contrast, cortical neurons derived from CRISPR/Cas9-corrected-iPSC lines showed clear IIH6 reactivity of the expected molecular size (~120 kDa), indicating restored glycosylation of α-dystroglycan (Fig 4B).

Following the detection of glycosylated α-dystroglycan in corrected cortical neurons, we then investigated whether the laminin-binding activity is associated with the IIH6 glyco-epitope on α-dystroglycan. We confirmed that laminin-binding was disrupted in cortical neurons derived from FKRP-iPSCs (Fig 4C), whereas cortical neurons derived from corrected-iPSC lines showed strong laminin-binding activity at ~120 kDa similar to mouse brain, but lower than mouse muscle (~156 kDa) (Fig 4C). Together, these results demonstrated a restoration of tissue-specific functional glycosylation of α-dystroglycan in CRISPR/Cas9-corrected cortical neurons.

### Targeted gene mutation of *FKRP* disrupts functional glycosylation of α-dystroglycan in cortical neurons derived from iPSCs

To independently validate the results from our FKRP- and corrected-iPSC-derived cortical neurons, the same genome-editing strategy was used to knock-in the *FKRP* c.1364C>A (p.A455D) mutation into the WT-iPSCs [49] (Appendix Fig S3A). A donor-targeting vector carrying the *FKRP* mutation for CRISPR/Cas9-mediated homologous recombination was constructed. We then electroporated the site-specific CRISPR/Cas9 plasmids with the targeting donor vector (*FKRP* c.1364C>A) into the WT-iPSCs (Appendix Fig S3B). Using puromycin-positive selection and PCR genotyping (Appendix Fig

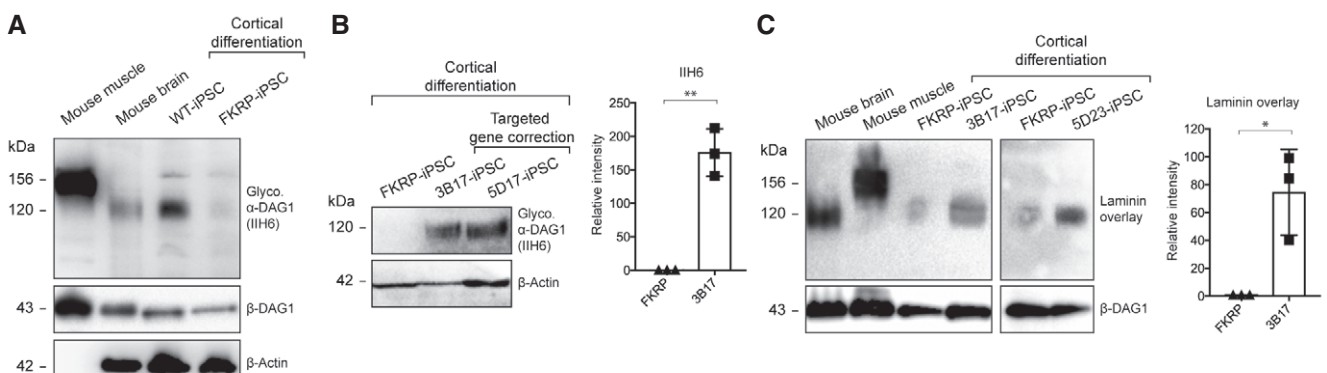

**Figure 4. Functional glycosylation of α-dystroglycan in cortical neurons derived from CRISPR/Cas9-corrected-iPSCs.**

A  Molecular weight of glycosylated α-dystroglycan (IIH6 epitope) in WT-iPSC-derived cortical neurons is similar to that in the mouse brain (~120 kDa) and lower than that in the mouse muscle (~156 kDa). IIH6 reactivity was almost not detected in cortical neurons derived from FKRP-iPSCs. Note that the β-actin antibody does not cross-react with the muscle sample.

B  Representative immunoblots show that targeted gene correction of *FKRP* restores IIH6 reactivity in iPSC-derived cortical neurons. Intensities of IIH6 reactivity are normalized to β-actin expression, indicating mean ± s.d. (n = 3 biological replicates; *t*-test; **P < 0.01).

C  Representative laminin overlays show that targeted gene correction restores laminin-binding activity in iPSC-derived cortical neurons. Intensities of laminin-binding activity are normalized to β-dystroglycan, indicating mean ± s.d. (n = 3 biological replicates; *t*-test; *P < 0.05).

S3C), 2 homozygously targeted clones (a15 and c43) were identified and expanded for selection cassette excision (Appendix Table S3). Following FIAU-negative selection and PCR genotyping (Appendix Fig S3C), we identified 2 biallelicly *FKRP* mutated-iPSC clones (C29 and C31) that have the selection cassette completely excised without re-integration (Appendix Table S3). The *FKRP* mutated-iPSC clones (C29 and C31) were sequenced to confirm the biallelic knock-in of *FKRP* c.1364C>A (p.A455D) mutations and the engineered selection cassette excision site (Appendix Fig S3D). The WT- and *FKRP* mutated-iPSC lines were differentiated to NSCs (Appendix Fig S4A and B) and subsequently to cortical neurons for functional analysis. Compared with cortical neurons derived from WT-iPSCs, targeted gene mutation of *FKRP* disrupted the IIH6 reactivity (Appendix Fig S4C) and laminin-binding activity (Appendix Fig S4D) in cortical neurons derived from *FKRP* mutated-iPSC lines. This indicated loss of functional glycosylation of α-dystroglycan. Together, our results demonstrated that target gene mutation of *FKRP* by CRISPR/Cas9-mediated genome editing could recapitulate pathological hallmarks of dystroglycanopathy in human iPSC-derived cortical neurons.

**High-throughput screening using H2K 2B4 mouse myoblast cell line for increased glycosylation of α-dystroglycan**

In parallel to generation of FKRP dystroglycanopathy human iPSC models, we sought to perform a high-throughput phenotypic screening to identify compounds capable of increasing the functional glycosylation of α-dystroglycan (detected by IIH6 immunostaining). Using an automated high-content microscopy platform, we screened a library of 31,954 chemical compounds with the immortalized mouse H2K 2B4 myoblast cell line in 384-well plates (Fig 5A). The compounds were screened at 10 μM and incubated with the H2K 2B4 cells for 24 h. Each plate also contained cells treated with 0.2% dimethyl sulfoxide (DMSO) as negative controls and cells incubated with 5 μM Trichostatin A [55] as positive controls. Following immunostaining and image acquisition, an automated image analysis algorithm was used to detect the fluorescent intensity of the IIH6 immunostaining. A total of 683 compounds (2.1%) were selected for further analysis (Fig 5B). Next, two independent inspectors

performed a series of visual examinations on algorithm-selected images. We excluded images with non-specific staining or autofluorescence and scored for the localization of IIH6 staining and percentage of IIH6-positive cells. From these, IIH6 fluorescent intensity values of the acquired images were compared to the negative and positive controls in the same plates and 11 compounds were assessed in a dose–response assay with seven different concentration points (Fig 5C). Three hits with highest increase in IIH6 immunostaining were pursued further, i.e., (5Z)-5-[(3-ethoxy-4-hydroxy-phenyl)methylene]-3-(4-fluorophenyl)-2-thioxo-thiazolidin-4-one (HPFPTzone) (Fig 5D, Structure 1), 4BPPNit (Fig 5D, Structure 2), and (5E)-5-[(2,5-dimethyl-1-phenyl-pyrrol-3-yl)methylene]-3-phenyl-2-thioxo-thiazolidin-4-one (DPPTzone) (Fig 5D, Structure 3). Next, flow cytometry analysis was used as the secondary assay to detect the IIH6-reactive glycans on H2K 2B4 cells treated with these three compounds, compared with untreated cells. 4BPPNit confirmed as hit by showing significantly increased mean fluorescence intensity of IIH6-reactive glycans, while there was no difference in the percentage of IIH6-positive cells (Fig 5E). HPFPTzone and DPPTzone were discarded for inconsistent results (data not shown). Therefore, 4BPPNit was the most promising hit compound.

**4BPPNit augments functional glycosylation of α-dystroglycan in human FKRP-iPSC-derived neural stem cells**

Next, we explored whether the FKRP dystroglycanopathy human iPSC models could be used to demonstrate restoration of functional glycosylation of α-dystroglycan by small molecule compounds. FKRP- and corrected-NSC lines were treated with the hit compound 4BPPNit (20 μM, based on the dose–response and flow cytometry analysis) for 24, 48, and 72 h and compared with DMSO only controls. There was no discernible cell toxicity between compound-treated and non-treated control cells. Immunoblotting with the IIH6 antibody detected glycosylated α-dystroglycan (~120 kDa) in corrected-NSCs (3B17), whereas the IIH6 reactivity was strongly reduced in FKRP-NSCs (Fig 6A). Compared with untreated control, increased IIH6 reactivity was detected in the 4BPPNit-treated FKRP-NSCs, but much weaker than that in CRISPR-corrected-NSCs

**Figure 5. High-throughput screen identifies compounds increasing glycosylation of α-dystroglycan.**

A  Schematic of the screening procedure. Compounds were dispensed in Matrigel-coated 384-well plates prior to seeding of murine myoblasts. Myoblasts were incubated for 24 h with 10 μM of the individual compounds, 5 μM of the positive control Trichostatin A or with the negative control DMSO. After fixation and staining with the IIH6 antibody, detecting glycosylated α-dystroglycan, the cells were imaged using the confocal PE Opera LX high-throughput microscope. The intensity of the α-dystroglycan glycosylation staining was determined, values normalized, and a hit list generated. Scale bar, 20 μm.

B  Compound validation strategy. After the primary screen, the α-dystroglycan glycosylation staining pattern of initially identified compounds was examined to exclude autofluorescent as well as toxic compounds. From the remaining compounds, 11 hit compounds were nominated after acquired images were compared to controls in the respective screening plate. A dose–response assay followed by flow cytometry assays further narrowed down hits to 1 compound.

C  Dose–response curve. H2K 2B4 cells were treated with serial dilutions of 11 hit compounds for 24 h and fixed, stained, and imaged as described above. The average values and standard deviations of the mean fluorescent intensities of technical triplicates were calculated and a non-linear regression curve was fitted for the three compounds showing the highest increase in α-dystroglycan glycosylation (IIH6 immunostaining). Representative images of cells treated with 5 μM of the respective compounds as well as EC50 values are shown. HPFPTzone: (5Z)-5-[(3-ethoxy-4-hydroxy-phenyl)methylene]-3-(4-fluorophenyl)-2-thioxo-thiazolidin-4-one. 4BPPNit: 4-(4-bromophenyl)-6-ethylsulfanyl-2-oxo-3,4-dihydro-1H-pyridine-5-carbonitrile. DPPTzone: (5E)-5-[(2,5-dimethyl-1-phenyl-pyrrol-3-yl)methylene]-3-phenyl-2-thioxo-thiazolidin-4-one. Regression curve and EC50 values were calculated using GraphPad Prism 7 software. In images shown, green represents glycosylated α-dystroglycan and blue Hoechst staining. Scale bar, 50 μm.

D  Structures of the 3 compounds showing the highest increase in α-dystroglycan glycosylation. Structure 1: HPFPTzone. Structure 2: 4BPPNit. Structure 3: DPPTzone.

E  Flow cytometry analysis confirmed that 4BPPNit treatment (20 μM) significantly increased IIH6-reactive glycan in mouse H2K 2B4 cells, compared with untreated cells. There was no significant difference in the percentage of IIH6-positive cells. Values indicate mean ± s.e.m. (n = 6, triplicates of 2 independent experiments; t-test; NS, not significant; **P < 0.01).

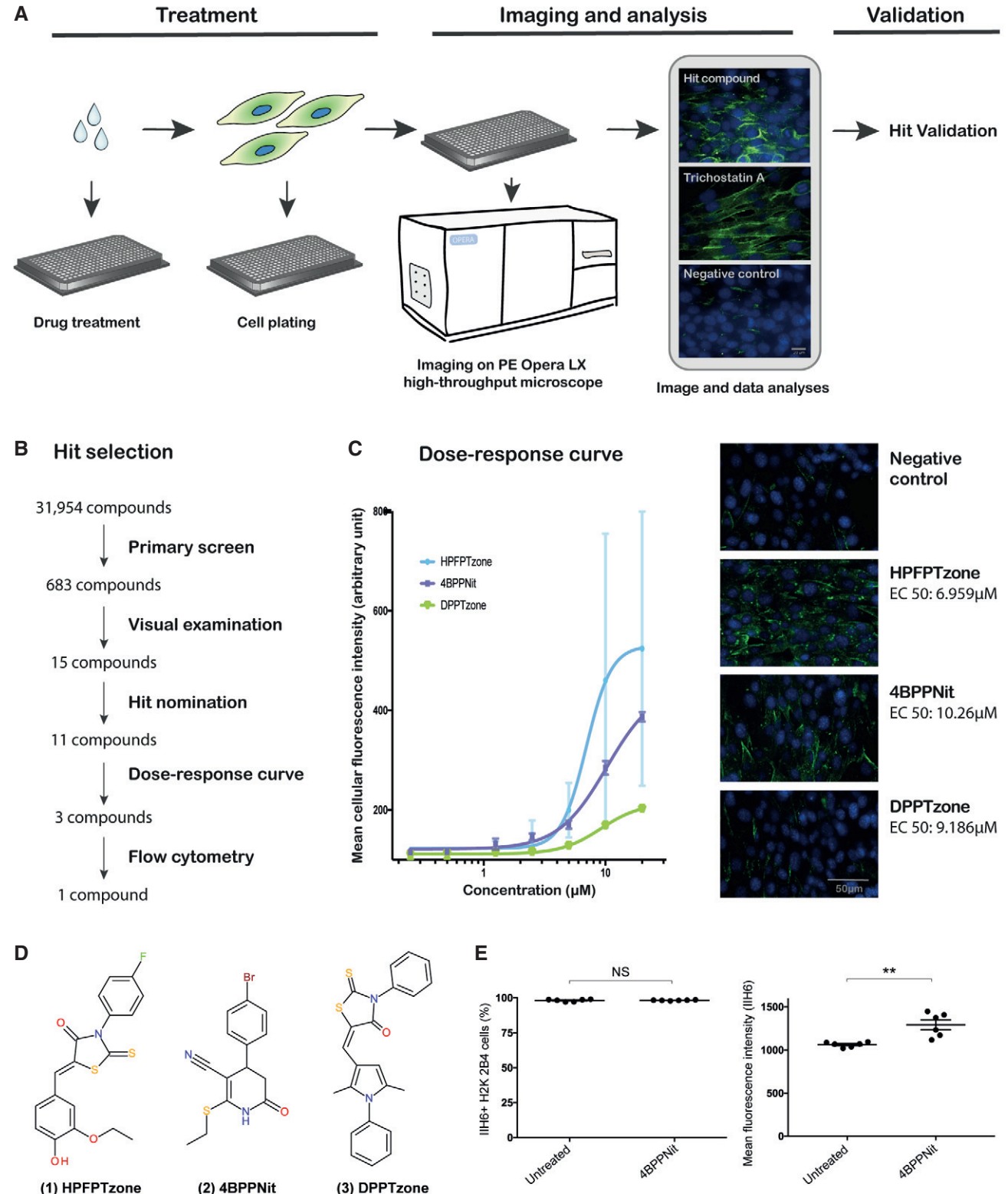

**Figure 5.**

(Fig 6A; Appendix Fig S5A). Under the same exposure time, the signal intensity of IIH6 reactivity is saturated in corrected-NSCs and its relative level is underestimated (Fig 6A). Thus, we decided to

quantify the effects of 4BPPNit treatment in FKRP-NSCs and CRISPR-corrected-NSCs separately (Fig 6B–G). FKRP-NSCs treated with 4BPPNit showed a significantly increased IIH6 reactivity

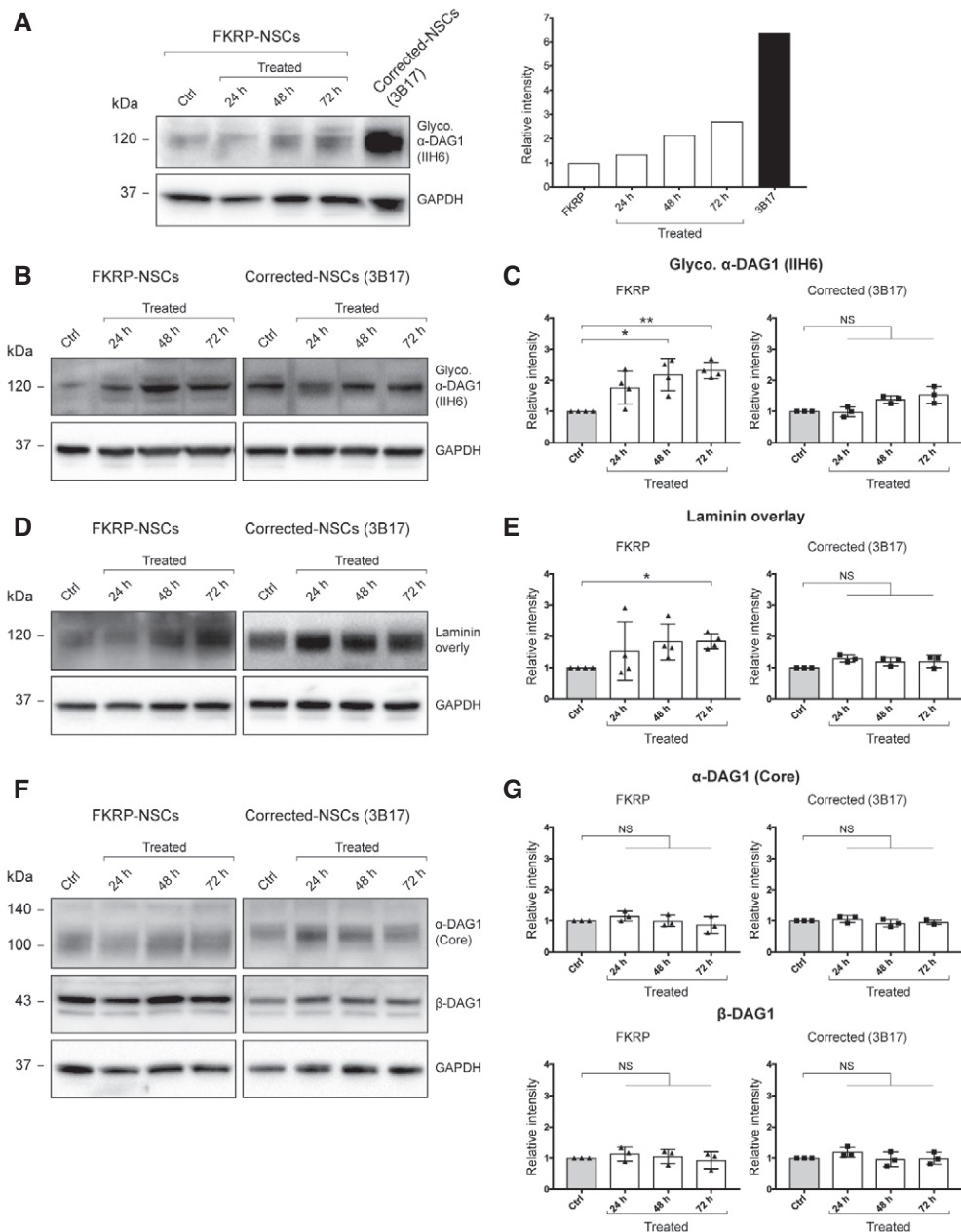

**Figure 6. Hit compound 4BPPNit validation in the FKRP dystroglycanopathy human iPSC models.**

A FKRP-NSCs treated with 4BPPNit showed increased IIH6 reactivity, compared with untreated control cells. Note that the augmented IIH6 reactivity by 4BPPNit is much weaker than that in CRISPR-corrected-NSCs. Under the same exposure time, the signal intensity of IIH6 reactivity is saturated in corrected-NSCs.

B Representative immunoblots with the IIH6 antibody using FKRP- and corrected-NSCs treated with 4BPPNit for 24, 48, and 72 h, and DMSO only controls.

C Quantification of α-dystroglycan glycosylation (IIH6), compared with DMSO only controls.

D Representative immunoblots of laminin-binding analysis using FKRP- and corrected-NSCs treated with 4BPPNit and DMSO only controls.

E Quantification of laminin-binding activities, compared with DMSO only controls.

F Representative immunoblots with the AF6868 antibody detecting both α-dystroglycan core protein and β-dystroglycan in FKRP- and corrected-NSCs treated with 4BPPNit and DMSO only controls.

G Quantification of α-dystroglycan core protein and β-dystroglycan expression, compared with DMSO only controls.

Data information: Intensities of glycosylation, laminin-binding activity, or protein expression are normalized to GAPDH protein expression. Note that the left and right panels of the immunoblots in (B) and (D) are not equivalent exposure time. See also Appendix Fig S5. Values indicate mean ± s.d. ($n$ = 3 or 4 biological replicates; one-way ANOVA; NS, not significant; *$P < 0.05$; **$P < 0.01$).

(~2-fold) at 48 and 72 h, compared with the untreated control (Fig 6B and C). A trend of slightly increased IIH6 reactivity in corrected-NSCs was observed, but this was not statistically significant (Fig 6B and C). Next, we confirmed significantly enhanced laminin-binding activity in FKRP-NSCs treated with 4BPPNit for 72 h, consistent with the increased IIH6 reactivity (Fig 6D and E). Treatment of 4BPPNit in corrected-NSCs increased the laminin-binding activity slightly, but this was not statistically significant (Fig 6D and E). Taken together, the increased IIH6 reactivity is correlated with enhanced laminin-binding activity in compound-treated FKRP-NSCs.

Next, we sought to distinguish whether the increased IIH6 reactivity and laminin-binding were due to increased glycosylation of α-dystroglycan or elevated dystroglycan core protein expression. To do this, we used a polyclonal antibody (AF6868) that recognizes both α-dystroglycan core protein and β-dystroglycan. Treatment of 4BPPNit at three time points did not significantly increase α-dystroglycan core protein or β-dystroglycan expression levels in FKRP-NSCs (Fig 6F and G). Similar results were observed in corrected-NSCs treated with 4BPPNit (Fig 6F and G). Note that the weak increase of IIH6 reactivity induced by 4BPPNit (Fig 6A; Appendix Fig S5A) was not sufficient to shift the molecular weight of α-dystroglycan core protein in FKRP-NSCs (Appendix Fig S5B). Together, these data indicate that 4BPPNit is capable of inducing tissue-specific functional glycosylation of α-dystroglycan in human FKRP-iPSC-derived NSCs, although the effect of 4BPPNit is much weaker than the CRISPR correction of *FKRP*.

### 4BPPNit induces upregulation of *LARGE1* gene expression

The function of 4BPPNit has not been reported previously. Studies have shown that overexpression of mutant *FKRP* can improve functional glycosylation of α-dystroglycan in FKRP mutant mice [56] and that overexpression of *LARGE1* in human cell lines can lead to enhanced functional glycosylation of α-dystroglycan [37]. In an attempt to address potential mechanisms by which 4BPPNit can induce functional glycosylation of α-dystroglycan, we hypothesized that 4BPPNit treatment may directly or indirectly upregulate gene expression levels of *FKRP* and/or *LARGE1* both of which are involved in synthesizing the core M3 glycan on α-dystroglycan. Consistent with immunoblotting analysis using the AF6868 antibody, dystroglycan (*DAG1*) gene expression was not significantly changed in either FKRP- or corrected-NSCs treated with 4BPPNit at three time points using quantitative real-time PCR (Fig 7A). Next, *FKRP* gene expression levels were measured in compound-treated FKRP- or corrected-NSCs. When compared with untreated cells, no significant differences were detected (Fig 7B). Interestingly, we found that 4BPPNit treatment significantly upregulated *LARGE1* gene expression levels in FKRP-NSCs treated for 48 h (~1.5-fold) and 72 h (~1.7-fold) and in corrected-NSCs treated for 72 h (~1.8-fold), when compared with untreated controls (Fig 7C; Appendix Fig S6). Analysis of other genes involved in the synthesis of core M3 glycan on α-dystroglycan showed a mild but significant increase of *B4GAT1*, *TMEM5*, *FKTN*, and *POMGNT2* expression in CRISPR-corrected-NSCs treated with 4BPPNit, but no significant difference was detected in FKRP-NSCs treated with 4BPPNit (Appendix Fig S6). Together, these results suggest that 4BPPNit treatment may increase functional glycosylation of α-dystroglycan in FKRP-NSCs through upregulation of *LARGE1* gene expression (see Discussion).

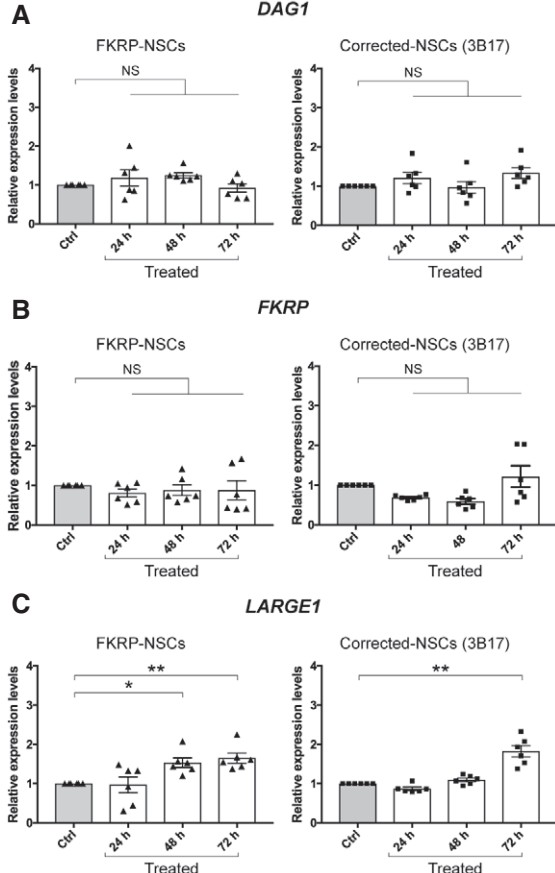

**Figure 7. Compound 4BPPNit induces upregulation of *LARGE1* gene expression.**

A  *DAG1* gene expression analysis in FKRP- and corrected-NSCs treated with 4BPPNit, compared with DMSO only controls.
B  *FKRP* gene expression analysis in FKRP- and corrected-NSCs treated with 4BPPNit, compared with DMSO only controls.
C  *LARGE1* gene expression analysis in FKRP- and corrected-NSCs treated with 4BPPNit, compared with DMSO only controls.

Data information: Gene expression levels are normalized to *ACTB* gene expression. Values indicate mean ± s.e.m. (*n* = 6 biological replicates; one-way ANOVA; NS, not significant; *$P < 0.05$; **$P < 0.01$).

## Discussion

In this study, we have developed a novel isogenic, human FKRP-iPSC-based platform for modeling dystroglycanopathy and testing potential drug candidates. We report here that CRISPR/Cas9-induced homologous recombination together with the *piggyBac* positive/negative selection cassette is a powerful and versatile strategy, which allows precise modification of the mammalian genome at a single base pair level without leaving footprints. For the first time, we demonstrate a precise genome editing at the *FKRP* locus in human iPSCs. Most reported *FKRP* pathogenic variants are missense mutations. Among these mutations, the *FKRP* c.826C>A (L276I) mutation is the most common variant and causes a mild form of dystroglycanopathy (LGMD2I), which is among the most common LGMD forms especially in the Caucasian population. Our genome-editing strategy can be applied to precisely correct these *FKRP*

missense mutations and by implication small insertion/deletion mutations (INDELs), as well as similar type of mutations in other dystroglycanopathy genes. Recent advances in the delivery of Cas9-sgRNA ribonucleoproteins (Cas9 RNPs) may further improve the efficiency of biallelic gene targeting [57].

The process of cellular reprogramming from fibroblasts to iPSCs followed by directed differentiation from iPSCs to NSCs and subsequently to cortical neurons involves dramatic changes to the gene expression profile of a cell. The functional glycosylation of α-dystroglycan is a consequence of a complex biosynthetic pathway orchestrated by many enzymes to form the laminin-binding moiety [5]. Our results indicate that genes involved in functional glycosylation of α-dystroglycan are expressed during *in vitro* corticogenesis. Interestingly, the molecular weight of α-dystroglycan varies due to tissue-specific *O*-glycosylation [1]. Consistent with previous studies, we have shown that the molecular weight of glycosylated α-dystroglycan in human iPSC-derived neural cells is similar to that in the mouse brain. Importantly, targeted gene correction of *FKRP* restores tissue-specific functional glycosylation of α-dystroglycan in iPSC-derived neurons, whereas targeted gene mutation of *FKRP* in unrelated wild-type cells disrupts this glycosylation. Our findings have significant implications in using human iPSCs for modeling dystroglycanopathies because the *in vitro* cellular model can recapitulate the *in vivo* pathological hallmarks of the target tissues.

Mutations in *FKRP* and other known causative genes together account for approximately 50-60% of severe forms of dystroglycanopathies [29,33]. Previously, a zebrafish-based study showed genetic interactions between *ISPD*, *FKTN,* and *FKRP* [29]. Subsequent studies demonstrated the enzymatic function of ISPD, FKTN, and FKRP, which synergistically add tandem ribitol-5-phosphate onto α-dystroglycan [24,25]. Furthermore, a recent study showed that excess amount of ribitol treatment in the FKRP(P448L) mutant mice can increase levels of ribitol-5-phosphate and CDP-ribitol, leading to enhanced functional glycosylation of α-dystroglycan associated with improved muscle pathology and function [58]. This therapeutic strategy is based on the residual enzymatic function of FKRP mutant proteins. In the present study, we showed that we have identified a hit compound 4BPPNit by high-throughput small molecule screening and demonstrated the use of FKRP dystroglycanopathy human iPSC-derived neural cells for validating this compound. Moreover, treatment of 4BPPNit can significantly induce upregulation of *LARGE1* gene expression in FKRP-NSCs, which is correlated with significantly increased functional glycosylation of α-dystroglycan. A combinatorial approach utilizing ribitol and 4BPPNit may have synergistic effects for augmenting functional glycosylation of α-dystroglycan. Interestingly, despite significant upregulation of *LARGE1* transcript in corrected-NSCs treated with 4BPPNit, this does not markedly enhance functional glycosylation of α-dystroglycan, suggesting a possible feedback mechanism for regulating the extension of xylose-glucuronic acid disaccharide repeats transferred by LARGE1 in the healthy condition.

We report here the first FKRP dystroglycanopathy human iPSC models and its suitability to assess functional glycosylation of α-dystroglycan using genetic and chemical correction. Although 4BPPNit treatment augmented functional glycosylation of α-dystroglycan (chemical correction), the effect is not sufficient to shift the molecular weight of α-dystroglycan core protein to normal levels as

in CRISPR-corrected cells (genetic correction). Future studies using our isogenic pairs of human iPSC models hold significant promise in early-stage drug discovery and for validation of potential small molecule drug candidates. Furthermore, it has been shown that downregulation of *LARGE1* and hypoglycosylation of α-dystroglycan are involved in cancer progression and metastasis [55,59]. Therefore, the identification of compounds with a similar mechanism of action as 4BPPNit may be applicable for developing cancer therapies.

Moreover, the isogenic pairs of human iPSCs generated in this study have enormous potential in other research avenues. Apart from degeneration of skeletal muscle, dilated cardiomyopathy is a frequent associated feature of FKRP-deficient dystroglycanopathy [60]. To study mechanisms underlying muscle degeneration and dilated cardiomyopathy, the isogenic pairs of corrected- and FKRP-iPSCs can be differentiated to myogenic and cardiac cells [61,62]. Importantly, this will facilitate drug discovery and development for treating a variety of diseases, in which loss of α-dystroglycan-laminin-binding activity underlies the disease pathogenesis.

# Materials and Methods

### Generation of human FKRP-iPSC lines

We obtained FKRP patient fibroblast lines from the MRC Centre for Neuromuscular Diseases Biobank (REC reference 06/Q0406/33) and ethical approval for the use of these cells (REC reference 13/LO/1826; IRAS project ID: 141100). All patients or their legal guardians gave written informed consent. To generate patient-specific induced pluripotent stem cells (iPSCs), we implemented an efficient reprogramming technology based on a doxycycline-inducible system using six factors, OCT4, SOX2, KLF4, c-MYC, RARG, and LRH1 [49]. *In vitro* differentiation and analysis were carried out as described [49].

### Vector construction and CRISPR/Cas9-mediated genome editing

For targeted gene correction, we used Gibson Assembly (New England BioLabs) to construct the targeting donor vector. Briefly, the 1-kb left and right homology arms (LHA and RHA) were PCR-amplified from parental *FKRP* fibroblast. The *FKRP* c.1364C>A (p.A455D) mutation on the LHA was simultaneously corrected with a modified primer (Appendix Table S4). The *piggyBac* (*PGK-puroΔtk*) selection cassette and vector backbone were PCR-amplified from the pMCS-AAT_PB-PGKpuroTK plasmid [63]. The four PCR fragments have 40-bp overlapping from end to end (primers are listed in Appendix Table S4) and joined together using Gibson Assembly Master Mix. The hCas9 plasmid was a gift from George Church (Addgene plasmid # 41815) [45]. Short DNA fragments containing the *FKRP* sgRNA target sequence (Appendix Table S1) were generated by annealing two primers (Appendix Table S4), which create two overhangs ready for cloning into BsaI-digested p1261_U6_BsaI_gRNA_plasmid (kind gift from Sebastian Gerety, Wellcome Trust Sanger Institute). The expression of sgRNA is under the control of U6 promoter. The cell suspension was transferred to a cuvette and electroporated using Amaxa Nucleofection 2b device with program B16. For targeted gene mutation, the same strategy was used to construct the donor-targeting vector carrying the *FKRP* c.1364C>A (p.A455D) mutation on LHA using primers in Appendix Table S4.

## Karyotyping analysis

As previously described [64], multiplex fluorescent *in situ* hybridization (M-FISH) karyotype analysis was performed on iPSC lines with slight modification. Briefly, prior to metaphase harvesting, iPSCs were grown in M15 medium (knockout DMEM, 15% fetal bovine serum, 1X glutamine–penicillin–streptomycin, 1X non-essential amino acids, and 1 ng/ml human recombinant LIF) for 24 h and then treated with 10 μM Y-27632 dihydrochloride (Tocris) for 2–3 h.

## Neural induction and cortical differentiation from human iPSCs

As described [52], induction of NSCs from human iPSCs was carried out using Gibco Neural Induction Medium (Neurobasal medium and 2% neural induction supplement) for 7 days, followed by passaging and expansion in neural expansion medium (49% Neurobasal medium, 49% Advanced DMDM/F-12 and 2% neural induction supplement). Expanded NSCs were cryopreserved or differentiated into cortical neurons following the established protocol [53]. Briefly, tissue culture plates were pre-coated with 0.01% (w/v) poly-L-ornithine (Sigma, P4957) for 4 h, followed by 20 μg/ml of laminins (Sigma, L2020) for 4 h. NSCs were seeded on the pre-coated plates (50,000/cm$^2$) in neural expansion medium with 5 μM Y-27632 dihydrochloride (Tocris) for 24 h. Subsequently, neural expansion medium was replaced with neural maintenance medium, containing 50% DMEM/F-12 (Gibco), 50% Neurobasal medium (Gibco), 0.5X N2 supplement (Gibco), 0.5X B27 supplement (Gibco), 1.5 mM GlutaMAX-I (Gibco), 0.5X penicillin–streptomycin (Gibco), 2.5 μg/ml insulin (Sigma), 0.05 mM of 2-mecaptoethanol (Gibco), 0.5% v/v of non-essential amino acid (Gibco), 20 ng/ml BDNF (Peprotech), and 20 ng/ml GDNF (Peprotech). When reaching 90% confluence (approximately within 3–4 days), cells were re-seeded (60,000/cm$^2$) in neural maintenance medium (with 5 μM Y-27632 dihydrochloride for 24 h), followed by media change every other day. *In vitro* corticogenesis occurs in the following weeks. Cultured cells were then harvested or fixed at specific time points for further analysis.

## Immunocytochemistry

Human iPSCs and iPSC-derived neural cells were fixed with 4% paraformaldehyde (PFA) for 15 min at room temperature (RT). Prior to immunocytochemistry, cells were permeabilized with 0.1% Triton/phosphate-buffered saline (PBS) for 15 min at RT and blocked with either 10% goat serum/PBS or 10% horse serum/PBS. Primary antibodies used were as follows: OCT4 (1:100; Santa Cruz, sc-5279), NANOG (1:100; Abcam, AB80892), Tra-1-60 (1:100; Santa Cruz, sc-21705), SSEA4 (1:100; BD Bioscience, 560796), α-fetoprotein (1:150; R&D Systems, MAB1368), α-smooth muscle actin (1:150; R&D Systems, MAB1420), Sox1 (1:100; R&D Systems, AF3369), Sox2 (1:100; R&D Systems, 245610), Nestin (1:500; Abcam, ab22035), Ki67 (1:100; BD Pharmingen, 550609), OTX2 (1:250; Millipore, AB9566), Pax6 (1:300; Biolegend, 901301), vimentin (1:100; Abcam, ab28028), vGLUT1 (1:2000; Synaptic systems, 135303), and β-III tubulin (Tuj1) (1:150; R&D System, MAB1195). All primary antibodies were diluted in blocking solution and incubated overnight at 4°C, followed by washing in PBS 3 times for 15 min. Subsequently, appropriate Alexa Fluor 488 or 564 conjugated secondary antibodies were incubated for 1 h at RT. For nuclei

staining, samples were incubated with DAPI (Sigma-Aldrich) for 5 min and washed in PBS briefly. Finally, images were captured and analyzed using IN Cell 2200 (GE Healthcare).

## Immunoblot analysis

Wild-type mouse muscle and brain lysates (kind gift from Susan Brown, Royal Veterinary College) were extracted as described [65]. Human iPSC-derived neural cells were harvested at specific time points, and proteins were extracted in RIPA buffer consisting of 50 mM Tris–HCl (pH 7.5), 150 mM NaCl, 1 mM EDTA, 1% Triton X-100, 1% SDS, and 1 mM azide plus a cocktail of protease inhibitors (Roche). A total of 30–60 μg of soluble protein was resolved using NuPage Novex 4-12% Bis–Tris protein gels (Invitrogen) and then electrophoretically transferred to polyvinylidene difluoride (PVDF) membrane (Millipore). The PVDF membrane was blocked in 3% bovine serum albumin in TBST (Tris-buffered saline with 0.1% Tween) and probed with primary antibodies to glycosylated α-dystroglycan (IIH6 1:500; kind gift from Kevin Campbell, University of Iowa), β-dystroglycan (1:100; Leica Biosystems, NCL-b-DG), and dystroglycan core protein (1:200; R&D Systems, AF6868), followed by washing with TBST and incubation with horseradish peroxidase (HRP)-conjugated anti-mouse IgM (Millipore), HRP-conjugated anti-mouse IgG (Jackson laboratories), and HRP-conjugated anti-sheep IgG (1:1000; R&D Systems, HAF016), respectively. Note that anti-beta actin antibody (1:1000; Abcam, ab8226) used in this study does not cross-react with skeletal muscle actin. SuperSignal West Pico Chemiluminescent Substrate (Thermo Fisher) was applied to PVDF membranes and signals were visualized using a Bio-Rad ChemiDoc MP imaging system and analyzed using ImageJ.

## Laminin overlay analysis

Protein samples were extracted, resolved, and transferred to PVDF membranes as described above. The PVDF membrane was then blocked with 5% skimmed milk in laminin-binding buffer (LBB), containing 10 mM triethanolamine, 140 mM NaCl, 1 mM MgCl$_2$, 1 mM CaCl$_2$ (pH adjusted to 7.6), followed by incubation with laminins from Engelbreth–Holm–Swarm murine sarcoma basement membrane (Sigma, L2020) in LBB with a final concentration of 5 μg/ml at 4°C overnight. The PVDF membrane was then incubated with the pan-laminin antibody (1:1000; Sigma, L9393), washed with 1× LBB with 0.1% Tween, and incubated with HRP-conjugated anti-rabbit IgG (Jackson laboratories). Signal detection using chemiluminescence substrate was performed as above.

## High-throughput compound screening

### Cell line, compound library, and platform

H2K 2B4 myoblasts were grown at 33°C in 10% $CO_2$ in medium containing γ-interferon (Chemicon) at a concentration of 20 U/ml. The flasks were coated with 0.1 mg/ml Matrigel (BD bioscience) diluted in DMEM (Gibco, Life Technology). The growth medium consists of 20% fetal bovine serum (FBS, Gibco), 2% L-glutamine (Sigma), and 1% penicillin/streptomycin (Gibco), plus 2% chick embryo extract (Sera lab) and γ-IFN (2 μl/ml, Merck). Large-scale screening was performed on 31,954 chemical compounds from the ChemiBank(UCL) collection (http://www.ucl.ac.uk/chemibank/)

[66]. Molecular properties ranges were as follows: molecular weight between 126 and 600, AlogP between −3.5 and 6, hydrogen bond donors between 0 and 6, hydrogen bond acceptors between 0 and 12, rotatable bonds between 0 and 15, and number of rings between 1 and 8. The library has hit-like properties but is outside of Lipinski's rule of five. Compounds were stored as a 10 mM DMSO stock solution under nitrogen (5% $O_2$) and low humidity (5%) at room temperature and in the dark (Roylan San Francisco storage pod). On the day of the screening, 384-well plates (BD bioscience) were coated with Matrigel and compounds were dispensed at the final concentration of 10 μM by an acoustic dispenser (Labcyte Echo 520). Cells were detached from their flasks using trypsin–EDTA (Life Technology), counted, diluted at the concentration of 10,000 cells/50 μl, and then plated in the coated 384 well plates. The media for the control wells (no compound added) contained 0.2% dimethyl sulfoxide (DMSO, Sigma) to account for the DMSO present in the compounds; as a positive control, cells were incubated with Trichostatin A (Sigma-Aldrich) 5 μM. After incubation for 24 h at 33°C, the cells were fixed with 2% PFA for 40 min and then washed using a Tecan HydroSpeed Plate Washer. The myoblasts were then incubated with anti-α-dystroglycan IIH6 (Millipore) for 1 h, followed by an anti-IgM-Alexa 488 (Life Technology) and Hoechst 33342 (Life Technology) for 40 min. Stained cells were imaged on a PerkinElmer Opera LX (a confocal high-content screening microscope) using the following light sources: UV Lamp (for Hoechst detection), 488 laser (for α-dystroglycan detection). Images were analyzed in Columbus 2.4.1 (Perkin Elmer) and the abundance of glycosylated α-dystroglycan was assessed. Briefly, cells were segmented and the mean whole-cell intensity was determined using the 488 Channel. The resulting intensity values were normalized to negative controls in each plate, and z-scores were calculated using the CellHTS2 program for R [67].

### Flow cytometry

H2K 2B4 cells were grown as described above in 75-$cm^2$ flasks. When the cells were approximately 80-90% confluent, they were treated with compounds at various concentrations for 24 h at 33°C/5% $CO_2$. After 24 h, cells were detached using non-enzymatic cell dissociation solution and fixed on ice with 2% paraformaldehyde for 10 min. Cells were washed with PBS ($Mg^{2+}$ and $Ca^{2+}$ free) and centrifuged and the pellets re-suspended with anti-α-dystroglycan IIH6 antibody (1:200 in 0.1% FBS/PBS). For a negative control, some cells were incubated without primary antibody and some with 0.2% dimethyl sulfoxide (DMSO, Sigma). Incubation was on ice for 30 min, followed by biotinylated anti-mouse IgM (1:100 in 0.1% FBS/PBS) for 20 min, and streptavidin-PE (1:1,000 in 0.1% FBS/PBS) for 15 min. Cells were then run on a Cyan ADP fitted with a 488 nm laser and a 633 nm red diode. The analysis was performed using FlowJo software.

Expanded View for this article is available online.

### Acknowledgements

We thank Dr Wei Wang (Wellcome Trust Sanger Institute) for his critical advice on the generation of human iPSCs. We are indebted to Dr Kosuke Yusa (Wellcome Trust Sanger Institute) for valuable discussion about genome editing and providing plasmids. We thank Dr Luke Gammon (Blizard Institute Screening Core Facility) for his kind assistance in some experiments. The support of the Muscular Dystrophy UK to the Dubowitz Neuromuscular Centre, of the MRC Neuromuscular Centre and of the NIHR Biomedical Research Centre at Great Ormond Street Hospital is also gratefully acknowledged. FM is supported by the National Institute for Health Research Biomedical Research Centre at Great Ormond Street Hospital for Children NHS Foundation Trust and University College London. This work was supported by Royal Society Research Grant (RG130417) and Newlife Research Grant (SG/14-15/14) to YYL, by Wellcome Trust Grant (098051) to SL, BF and FY, and by EU FP7 and EFPIA (n° 115582) and NEUROMICS (n° 2012-305121) to FM. This work was also supported by the Medical Research Council Core funding to the MRC LMCB (MC_U12266B) and MRC Capital Investment Fund (92-963).

## Author contributions

JK, BL, DR, EK, and AP-R carried out all the iPSC related work, analyzed, and interpreted the data. ST, FC, PA, CL, and ES performed compound screens. SL, BF, and FY performed the karyotyping analysis. AWEC performed cheminformatics. DR, DLSt, PL, ST, RK, DLSe, and FM contributed to the conception and design of the project and edited the manuscript. Y-YL designed the project, performed some of the experiments, analyzed and interpreted the data, and supervised the research. JK, FM, and Y-YL co-wrote the manuscript.

## Conflict of interest

The authors declare that they have no conflict of interest.

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
