## [Review Process File · EMBO Reports]

A new patient derived iPSC model for dystroglycanopathies validates a compound that increases glycosylation of α -dystroglycan

Jihee Kim, Beatrice Lana, Silvia Torelli, David Ryan, Francesco Catapano, Pierpaolo Ala, Christin Luft, Elizabeth Stevens, Evangelos Konstantinidis, Sandra Louzada, Beiyuan Fu, Amaia Paredes-Redondo, A. W. Edith Chan, Fengtang Yang, Derek L. Stemple, Pentao Liu, Robin Ketteler, David L. Selwood, Francesco Muntoni, Yung-Yao Lin

Review timeline:

Submission date:	4 March 2019
Editorial Decision:	29 March 2019
Revision received:	2 July 2019
Editorial Decision:	2 August 2019
Revision received:	24 August 2019
Accepted:	29 August 2019

Editor: Deniz Senyilmaz-Tiebe

Transaction Report:

1st Editorial Decision

29 March 2019

Thank you for submitting your manuscript for consideration by EMBO Reports. It has now been seen by three referees whose comments are shown below.

As you can see, all referees express interest in the presented FKRP model and the newly identified drug that increases glycosylation levels in this model. However, they also raise concerns that need to be addressed in full before we can consider publication of the manuscript here. In particular, the referees would like to have more insight into the upregulation of glycosylation by 4BPPN in the FKRP model.

Given these constructive comments, I would like to invite you to revise your manuscript with the understanding that the referee must be fully addressed and their suggestions taken on board. Please address all referee concerns in a complete point-by-point response. Acceptance of the manuscript will depend on a positive outcome of a second round of review. It is EMBO Reports policy to allow a single round of revision only and acceptance or rejection of the manuscript will therefore depend on the completeness of your responses included in the next, final version of the manuscript.

You can submit the revision either as a Scientific Report or as a Research Article. For Scientific

Reports, the revised manuscript can contain up to 5 main figures and 5 Expanded View figures. If the revision leads to a manuscript with more than 5 main figures it will be published as a Research Article. If a Scientific Report is submitted, these sections have to be combined. This will help to shorten the manuscript text by eliminating some redundancy that is inevitable when discussing the same experiments twice. In either case, all materials and methods should be included in the main manuscript file.

Supplementary/additional data: The Expanded View format, which will be displayed in the main HTML of the paper in a collapsible format, has replaced the Supplementary information. You can submit up to 5 images as Expanded View. Please follow the nomenclature Figure EV1, Figure EV2 etc. The figure legend for these should be included in the main manuscript document file in a section called Expanded View Figure Legends after the main Figure Legends section. Additional Supplementary material should be supplied as a single pdf labeled Appendix. The Appendix includes a table of content on the first page with page numbers, all figures and their legends. Please follow the nomenclature Appendix Figure Sx throughout the text and also label the figures according to this nomenclature. For more details please refer to our guide to authors.

When preparing your letter of response to the referees' comments, please bear in mind that this will form part of the Review Process File, and will therefore be available online to the community. For more details on our Transparent Editorial Process, please visit our website: http://emboj.embopress.org/about#Transparent_Process

Regarding data quantification, please ensure to specify the name of the statistical test used to generate error bars and P values, the number (n) of independent experiments underlying each data point (not replicate measures of one sample), and the test used to calculate p-values in each figure legend. Discussion of statistical methodology can be reported in the materials and methods section, but figure legends should contain a basic description of n, P and the test applied. Please also include scale bars in all microscopy images.

We now strongly encourage the publication of original source data with the aim of making primary data more accessible and transparent to the reader. The source data will be published in a separate source data file online along with the accepted manuscript and will be linked to the relevant figure. If you would like to use this opportunity, please submit the source data (for example scans of entire gels or blots, data points of graphs in an excel sheet, additional images, etc.) of your key experiments together with the revised manuscript. Please include size markers for scans of entire gels, label the scans with figure and panel number, and send one PDF file per figure.

- a complete author checklist, which you can download from our author guidelines (<http://emboj.embopress.org/authorguide#revision>). Please insert page numbers in the checklist to indicate where the requested information can be found.
 - a letter detailing your responses to the referee comments in Word format (.doc)
 - a Microsoft Word file (.doc) of the revised manuscript text
 - editable TIFF or EPS-formatted figure files in high resolution
- (In order to avoid delays later in the publication process please check our figure guidelines before preparing the figures for your manuscript:
http://www.embopress.org/sites/default/files/EMBOPress_Figure_Guidelines_061115.pdf)
- a separate PDF file of any Supplementary information (in its final format)
 - all corresponding authors are required to provide an ORCID ID for their name. Please find instructions on how to link your ORCID ID to your account in our manuscript tracking system in our Author guidelines (<http://emboj.embopress.org/authorguide>).

As part of the EMBO publication's Transparent Editorial Process, EMBO reports publishes online a Review Process File to accompany accepted manuscripts. This File will be published in conjunction with your paper and will include the referee reports, your point-by-point response and all pertinent correspondence relating to the manuscript.

I look forward to seeing a revised version of your manuscript when it is ready. Please let me know if you have questions or comments regarding the revision.

REFEREE REPORTS

Referee #1:

In the presented paper, the authors generate an iPSC model system(s) of FKRP deficiency that (when differentiated into the neuronal lineage) faithfully recapitulates the dystroglycan glycosylation pattern commonly observed in brain. They also screen for small molecular compounds that increase α -dystroglycan glycosylation in a murine myoblast cell line and validate that one of the hit compounds increases glycosylation of α -dystroglycan in their FKRP-deficient iPSC system (likely in part through an up regulation of LARGE). Thus, they have generated an experimental system that can be used to test future therapies for FKRP-deficient therapies in a manner that correctly takes into account the brain-specific glycosylation pattern of α -dystroglycan. The data presentation is clear and the paper is well written.

While the paper does not add fundamentally new mechanistic insights into dystroglycanopathies, the generation of this experimental system and the generation of a functional hit compound are very much appreciated by this referee, in particular since comparable gene targeting approaches could be used to generate comparable experimental models for other dystroglycanopathies.

Comments:

1. Glycosylation of α -dystroglycan is a multi-step process. The authors demonstrate that treatment with 4BPPNt increases LARGE (but leaves FKRP unchanged). Given the rather modest increase in LARGE expression (which does not really precede the increase of glycosylation), this referee would have been interested to see whether transcript levels of other parts of the synthetic machinery (TMEM5, B4GAT, LARGE2) might also be increased by their hit compound.

2. This referee finds it difficult to judge whether a 2 fold increase of glycosylation is "a lot or not". My reasoning is as follows: In Figure 6B and 6D, the authors convincingly show that functional glycosylation in FKRP-mutant cells increases by approximately 2x upon treatment with 4BPPNt. In these figures, the "relative" intensity in the mutant and corrected cells is always normalised the intensity in untreated cells, so it is difficult to judge what percentage of the normal glycosylation levels is achieved upon treatment of the mutant cells. Quantification data in Figure 4B and C seems to indicate that IIIH6 staining and laminin overlay signal is approximately 100x lower in mutants than in wild type iPSCs. Thus, I would conclude that the 2-fold increase in figure 6 corresponds to an increase from 1% of the normal staining to 2% of the normal staining intensity. Yet, looking at the IIIH6 blots and the laminin overlay blots (and assuming that the exposure times between the left and right panels of fig 6A and 6C are comparable), I get the impression that the level obtained in the treated mutant cell lines is more than 2% of the normal levels. Thus to me, the quantitative presentation in Figure 6 (when seen in light of the results of Figure 4) might belittle the effects that were observed. I am therefore wondering whether it would be possible to present the quantitative data in both panels of Figure 6B and 6D, normalised the the intensity observed in untreated "corrected" cells.

Referee #2:

This report describes the generation of isogenic iPSC-derived neuronal lineages to study glycosylation defects of α -dystroglycan caused by a homozygous mutation in the FKRP gene. These isogenic lines were derived from a patient carrying a biallelic p.A455D missense mutation and CRISPR/CAS9 mutation-corrected derivatives thereof. Mutant neuronal cells showed clear

disruption of alpha-dystroglycan glycosylation, which was normalized in the isogenic corrected neuronal cells. In addition, the glycosylation defect could also be rescued by a compound, 4BPPNit, which was identified upon screening of a library of 31.954 compounds using dystroglycan glycosylation of a mouse myoblast line as readout. Finally, the authors show that the application of 4BPPNit is associated with an increase expression of LARGE, a glycosyltransferase that acts on building of the same glycan structure on dystroglycan as FKRP. The generation of FKRP mutant and wt isogenic pairs is an important achievement, providing proof-of-concept for a strategy that is likely to be followed for many other types of dystroglycanopathies and even other genetic disorders. For dystroglycanopathies numerous mutations affecting at least 18 genes are known. Isogenic iPSC pairs as developed here will have high utility for diagnostics, i.e. the assessment of variants of unknown significance identified upon exome/genome sequencing. In addition, the authors have shown the significant value of their strategy for fundamental research as well as identification and preclinical validation of compounds with therapeutic potential.

That said, I have some comments that need some attention:

1. Like many other glycosylation enzymes that underlie dystroglycanopathies, mutations give rise to a very broad range of clinical severity of the phenotype. FKRP is one of the most commonly involved genes in these disorders, and as indicated by the authors, this is especially the case for a single variant, p.L276I, which underlies a relative mild and late onset phenotype, Limb girdle muscular dystrophy type 2I (LGMD2I). Therefore, it is highly surprising that the authors have chosen an extremely rare (almost unique) variant to provide their proof of concept. What is the reason for that? The clinical relevance of this the LGMDI is so much higher. Proof of concept for this variant (or perhaps another one such as a complete loss-of-function in addition to the chosen rare variant) would have strengthened this study, by providing stronger proof-of-concept and possibly by providing more mechanistic insight into the points mentioned under 2 and 3.
2. IIH6 staining (against α -dystroglycan M3 glycan) and laminin overlay assay show that α -dystroglycan is strongly disrupted in mutant iPSC-derived neurons. However, this disruption is certainly not complete as residual bands are seen in figure 4 and in figure 6 these bands are even quite strong. In fact, panel 6E shows a almost normal amount of dystroglycan protein migrating at the same height of normally glycosylated dystroglycan. Apparently, the mutation under investigation is not a full loss-of-function and the effect of this hypomorphic allele appears variable under the conditions that were investigated. This notion has some consequences regarding the conclusion that FKRP is not required for neural differentiation (line 35, page 6).
3. The positive effect of 4BPPNit was clearly established in the elegant screen using H2K 2B4 mouse myoblasts. These cells express IIH6-positive dystroglycan. The augmentation of functional glycosylation was subsequently shown in mutant-iPSC-derived neural cells, which also express variable amounts of glycosylated α -dystroglycan as mentioned above. Probably, the presence of (residual) glycosylated protein is a prerequisite for the effect. However, I am not convinced that this is due to up-regulation of LARGE. To me, this is the weakest part of the report, because there is no direct evidence presented. What is the mechanism of up-regulation? Is it specific for LARGE or are other genes upregulated?, is there also upregulation of LARGE protein (only qRT-PCR was done (fig 7), not a WB as in other figures)?, why not a dose-response study? In addition, In figure 6E I do not see any evidence for an increased MW of dystroglycan protein in treated cells, would that not be expected if increased LARGE activity is the underlying mechanism?

Referee #3:

This is a highly focused, self-contained paper that describes the construction of a single patient-derived iPSC cell line, an isogenic mutation-corrected iPSC line, and the a control line into which the specific mutation was introduced. Most of the study was focused on documenting the well done experiments and describing the approaches in fine detail. These cells are differentiated along a neurological pathway. Together this is valuable, worthwhile and technically significant accomplishment. The conclusions are well justified.

They also screened a compound library for species that restore laminin binding to FKRP-deficient differentiated iPSC cells. However, they employed normal muscle cells for screening, not FKRP-deficient iPSC cells. They found one compound that increased LARGE expression 2-fold in both normal and deficient cells. It would be much better if they used FKRP-deficient cells. That choice and the outcome decreased my enthusiasm. LARGE is already known to be a limiting step, so finding a drug that up-regulated LARGE was almost inevitable using normal cells. I suggest that the

authors to explain that choice.

1st Revision - authors' response

2 July 2019

Response to referee comments

Referee #1:

In the presented paper, the authors generate an iPSC model system(s) of FKRP deficiency that (when differentiated into the neuronal lineage) faithfully recapitulates the dystroglycan glycosylation pattern commonly observed in brain. They also screen for small molecular compounds that increase α -dystroglycan glycosylation in a murine myoblast cell line and validate that one of the hit compounds increases glycosylation of α -dystroglycan in their FKRP-deficient iPSC system (likely in part through an up regulation of LARGE). Thus, they have generated an experimental system that can be used to test future therapies for FKRP-deficient therapies in a manner that correctly takes into account the brain-specific glycosylation pattern of α -dystroglycan. The data presentation is clear and the paper is well written.

While the paper does not add fundamentally new mechanistic insights into dystroglycanopathies, the generation of this experimental system and the generation of a functional hit compound are very much appreciated by this referee, in particular since comparable gene targeting approaches could be used to generate comparable experimental models for other dystroglycanopathies.

We acknowledge the positive comments from Referee #1.

Comments:

1. Glycosylation of α -dystroglycan is a multi-step process. The authors demonstrate that treatment with 4BPPNit increases LARGE (but leaves FKRP unchanged). Given the rather modest increase in LARGE expression (which does not really precede the increase of glycosylation), this referee would have been interested to see whether transcript levels of other parts of the synthetic machinery (TMEM5, B4GAT, LARGE2) might also be increased by their hit compound.

The referee is correct in pointing out that dystroglycan glycosylation is a common multistep process. In addition to the characterisation of *LARGE1*, *FKRP* and *DAG1* expression, we have now performed qPCR experiments to detect transcript expression levels of other genes involved in the synthesis of core M3 glycan on α -dystroglycan, including *B4GAT1*, *TMEM5*, *ISPD*, *FKTN* and *POMGNT2*. We have added the results in Appendix Figure S6.

Briefly, except *ISPD* expression, *B4GAT1*, *TMEM5*, *FKTN* and *POMGNT2* expression was significantly upregulated ~ 1.5 fold in corrected-NSCs treated with 4BPPNit, compared with untreated control cells. However, no significant differences in expression levels of these genes were detected in FKRP-NSCs treated with 4BPPNit. Together, these results suggest that 4BPPNit may modulate the glycosylation pathway of α -dystroglycan, although none of the genes studied, apart from *LARGE1*, appear to be associated with the increased glycosylation in the mutant cell line. These data suggest that further investigation on the mechanism of action of 4BPPNit will be required, although they are beyond the scope of this manuscript.

In addition, and according to previous studies (e.g. HPA and GTEx datasets), *LARGE2* transcript is not expressed in the cerebral cortex. Consistent with this, we could not detect *LARGE2* expression in our human iPSC-derived NSCs by qPCR.

2. This referee finds it difficult to judge whether a 2 fold increase of glycosylation is "a lot or not". My reasoning is as follows: In Figure 6B and 6D, the authors convincingly show that functional glycosylation in FKRP-mutant cells increases by approximately 2x upon treatment with 4BPPNit. In these figures, the "relative" intensity in the mutant and corrected cells is always normalised the intensity in untreated cells, so it is difficult to judge what percentage of the normal glycosylation levels is achieved upon treatment of the mutant cells. Quantification data in Figure 4B and C seems

to indicated that IIH6 staining and laminin overlay signal is approximately 100x lower in mutants than in wild type iPSCs. Thus, I would conclude that the 2-fold increase in figure 6 corresponds to an increase from 1% of the normal staining to 2 % of the normal staining intensity. Yet, looking at the IIH6 blots and the laminin overlay blots (and assuming that the exposure times between the left and right panels of fig 6A and 6C are comparable), I get the impression that the level obtained in the treated mutant cell lines is more than 2% of the normal levels. Thus to me, the quantitative presentation in Figure 6 (when seen in light of the results of Figure 4) might belittle the effects that were observed. I am therefore wondering whether it would be possible to present the quantitative data in both panels of Figure 6B and 6D, normalised the the intensity observed in untreated "corrected" cells.

In Figure 6, immunoblot images in the left and right panel were not acquired with the same exposure time. As the referee noted, the residual IIH6-binding activity and laminin overlay in the *FKRP* mutant cells were much less than those in the CRISPR-corrected cells (Figure 4). When quantifying the IIH6 and laminin overlay signal intensity in cells treated or untreated with the compound 4BPPNit (Figure 6), we had to measure separately the *FKRP* mutant cell set and the CRISPR-corrected cell set. This is because genetic correction by CRISPR completely restored the IIH6-binding glycan and resulted in very strong signal intensity on immunoblots, whereas the chemical correction by 4BPPNit is not as complete as the genetic correction.

In order to see the effect of 4BPPNit in *FKRP* mutant cells, the immunoblots required longer exposure time. The signal intensity of IIH6 in 4BPPNit-treated cells was normalised to untreated cells. Longer exposure time for CRISPR-corrected cells would cause saturated signal intensity for IIH6 and laminin overlay on the immunoblots. To show this, we have added representative images of uncropped immunoblots in Appendix Figure S5. We also have added in the Figure 6 legend the following sentence: "Note that the left and right panels of the immunoblots in (A) and (C) are not equivalent exposure time. See also Appendix Fig. S5."

Referee #2:

This report describes the generation of isogenic iPSC-derived neuronal lineages to study glycosylations defects of alpha-dystroglycan caused by a homozygous mutation in the *FKRP* gene. These isogenic lines were derived from a patient carrying a biallelic p.A455D missense mutation and CRISPR/CAS9 mutation-corrected derivatives thereof. Mutant neuronal cells showed clear disruption of alpha-dystroglycan glycosylation, which was normalized in the isogenic corrected neuronal cells. In addition, the glycosylation defect could also be rescued by a compound, 4BPPNit, which was identified upon screening of a library of 31.954 compounds using dystroglycan glycosylation of a mouse myoblast line as readout. Finally, the authors show that the application of 4BPPNit is associated with an increase expression of *LARGE*, a glycosyltransferase that acts on building of the same glycan structure on dystroglycan as *FKRP*. The generation of *FKRP* mutant and wt isogenic pairs is an important achievement, providing proof-of-concept for a strategy that is likely to be followed for many other types of dystroglycanopathies and even other genetic disorders. For dystroglycanopathies numerous mutations affecting at least 18 genes are known. Isogenic iPSC pairs as developed here will have high utility for diagnostics, i.e. the assessment of variants of unknown significance identified upon exome/genome sequencing. In addition, the authors have shown the significant value of their strategy for fundamental research as well as identification and preclinical validation of compounds with therapeutic potential.

We thank Referee #2 for recognising the significant value of our human *FKRP* dystroglycanopathy iPSC models, which represents the first iPSC model of this genotype developed so far.

That said, I have some comments that need some attention:

1. Like many other glycosylation enzymes that underlie dystroglycanopathies, mutations give rise to a very broad range of clinical severity of the phenotype. *FKRP* is one of the most commonly involved genes in these disorders, and as indicated by the authors, this is especially the case for a single variant, p.L276I, which underlies a relative mild and late onset phenotype, Limb girdle muscular dystrophy type 2I (LGMD2I). Therefore, it is highly surprising that the authors have chosen an extremely rare (almost unique) variant to provide their proof of concept. What is the

reason for that? The clinical relevance of this the LGMDI is so much higher. Proof of concept for this variant (or perhaps another one such as a complete loss-of-function in addition to the chosen rare variant) would have strengthened this study, by providing stronger proof-of-concept and possibly by providing more mechanistic insight into the points mentioned under 2 and 3.

As we specified in the Introduction, there is an unmet need for physiology-relevant human cell models to test potential drug candidates in a neural-specific context. For this proof-of-concept study, we chose the FKRP(A455D) variant over the FKRP(L276I) variant because patients with the FKRP(A455D) variant exhibit both congenital muscular dystrophy and central nervous system involvement (Louhichi et al., 2004). In contrast, patients with the FKRP(L276I) variant show only mild and late onset muscle phenotype without brain involvement.

We agree with the referee that it will be interesting to compare different *FKRP* variants, such as a complete loss-of-function mutation and the FKRP(L276I) variant. Nevertheless, this will require differentiation of iPSCs to myogenic culture, which is outside the scope of this study.

We have added the following reference in the manuscript:

Louhichi et al. (2004) New FKRP mutations causing congenital muscular dystrophy associated with mental retardation and central nervous system abnormalities. Identification of a founder mutation in Tunisian families. *Neurogenetics* 5: 27–34.

2. IIH6 staining (against α -dystroglycan M3 glycan) and laminin overlay assay show that α -dystroglycan is strongly disrupted in mutant iPSC-derived neurons. However, this disruption is certainly not complete as residual bands are seen in figure 4 and in figure 6 these bands are even quite strong. In fact, panel 6E shows a almost normal amount of dystroglycan protein migrating at the same height of normally glycosylated dystroglycan. Apparently, the mutation under investigation is not a full loss-of-function and the effect of this hypomorphic allele appears variable under the conditions that were investigated. This notion has some consequences regarding the conclusion that FKRP is not required for neural differentiation (line 35, page 6).

Compared with the controls, the IIH6 reactivity and laminin overlay of FKRP(A455D) are indeed strongly reduced (Figure 4). When longer exposure time is applied to immunoblots, we could detect the residual signal of IIH6 reactivity and laminin overlay in FKRP(A455D) cells.

In fact, the molecular weight (MW) of dystroglycan core protein in FKRP(A455D) cells (Figure 6E, left panel) is lower than that in CRISPR-corrected cells (Figure 6E, right panel): the FKRP cells are ~100 kDa, whereas CRISPR-corrected cells are above 100 kDa. Note that the MW of glycosylated α -dystroglycan in the brain is ~120 kDa, which is much less than that in the muscle (~156 kDa). Thus, the migration of dystroglycan core protein does not seem to be big difference between FKRP and CRISPR-corrected neural cells. To clarify this, we have added an uncropped immunoblot in Appendix Figure S5B to show the migration difference of α -dystroglycan core proteins between FKRP and CRISPR-corrected samples.

Although we cannot exclude that FKRP(A455D) may have residual enzymatic activity, Table 1 clearly documents the patient's severe clinical phenotypes, suggesting the function of FKRP(A455D) is severely disrupted.

Regarding the conclusion that “FKRP is not required for neural differentiation”, we agree it may be misleading. To make it clearer, we have changed this sentence to read as “Although the FKRP(A455D) variant does not appear to be required during early neurogenesis, we cannot exclude that this mutation might be responsible for pathological phenotypes at a later stage.”

3. The positive effect of 4BPPNIt was clearly established in the elegant screen using H2K 2B4 mouse myoblasts. These cells express IIH6-positive dystroglycan. The augmentation of functional glycosylation was subsequently shown in mutant-iPSC-derived neural cells, which also express variable amounts of glycosylated α -dystroglycan as mentioned above. Probably, the presence of (residual) glycosylated protein is a prerequisite for the effect. However, I am not convinced that this is due to up-regulation of LARGE. To me, this is the weakest part of the report, because there is no direct evidence presented. What is the mechanism of up-regulation? Is it specific for LARGE or are other genes upregulated?, is there also upregulation of LARGE protein (only qRT-PCR was done

(fig 7), not a WB as in other figures)?, why not a dose-response study? In addition, In figure 6E I do not see any evidence for an increased MW of dystroglycan protein in treated cells, would that not be expected if increased LARGE activity is the underlying mechanism?

We agree with Referee #2 that the residual functions of FKRP(A455D) variant is critical for the effect of chemical correction by 4BPPNIt. We already performed a dose-response assay for 4BPPNIt, which has EC50: 10.26 μ M (Figure 5C). We have done additional qPCR experiments with 4BPPNIt at higher dose and detected *LARGE1* gene expression. We have added the results in Appendix Fig. S6. We show that *LARGE1* expression is significantly upregulated ~2 fold in FKRP- and corrected-NSCs treated with either 20 or 60 μ M 4BPPNIt for 72 hours, compared with DMSO only controls. No significant difference between 20 and 60 μ M 4BPPNIt-treated NSCs (Appendix Fig. S6A and G).

We also investigated the transcript levels of other genes involved in the synthesis of core M3 glycan on α -dystroglycan (Appendix Figure S6). In the main text, we added the following sentence to describe the results: “Analysis of other genes involved in the synthesis of core M3 glycan on α -dystroglycan showed a mild but significant increase of *B4GAT1*, *TMEM5*, *FKTN* and *POMGNT2* expression in CRISPR-corrected NSCs treated with 4BPPNIt, but no significant difference was detected in FKRP-NSCs treated with 4BPPNIt (Appendix Figure S6).”

We also studied protein expression of *LARGE1* with an antibody (proteintech, 24307-1-AP). However, we were not convinced that this antibody was capable of detecting the endogenous *LARGE1* protein. We agree with the referee that it will require comprehensive investigation regarding the mechanisms of *LARGE1* up-regulation induced by 4BPPNIt. Our study provides for the first time a platform with which FKRP mutant iPSCs could be studied in the future.

Although it is expected to see increased MW of α -dystroglycan core protein when functional glycosylation is restored, 4BPPNIt-treated FKRP cells have modest increase of IIH6 reactivity (Appendix Fig. S5A) and thus, the MW of α -dystroglycan core protein in 4BPPNIt-treated FKRP-cells does not shift to the same MW level as the α -dystroglycan core protein in CRISPR-corrected cells (Appendix Fig. S5B). To clarify this point, we have added the following sentence in the main text to read “Note that the modest increase of IIH6 reactivity induced by 4BPPNIt (Appendix Fig. S5A) was not sufficient to shift the molecular weight of α -dystroglycan core protein in FKRP-NSCs (Appendix Fig. S5B).”

Referee #3:

This is a highly focused, self-contained paper that describes the construction of a single patient-derived iPS cell line, an isogenic mutation-corrected iPS line, and the a control line into which the specific mutation was introduced. Most of the study was focused on documenting the well done experiments and describing the approaches in fine detail. These cells are differentiated along a neurological pathway. Together this is valuable, worthwhile and technically significant accomplishment. The conclusions are well justified.

We thank Referee #3 for appreciating the importance of our study.

They also screened a compound library for species that restore laminin binding to FKRP-deficient differentiated iPS cells. However, they employed normal muscle cells for screening, not FKRP-deficient iPS cells. They found one compound that increased *LARGE* expression 2-fold in both normal and deficient cells. It would be much better if they used FKRP-deficient cells. That choice and the outcome decreased my enthusiasm. *LARGE* is already known to be a limiting step, so finding a drug that up-regulated *LARGE* was almost inevitable using normal cells. I suggest that the authors to explain that choice.

We agree with Referee #3 that an important step could also be to perform a compound screen with FKRP-deficient cells. Nevertheless, a tight control of the conditions for a high-throughput screening is an imperative need when performing these studies. In view of the complexity of dystroglycan glycosylation during different stages of development, we elected to perform the original large-scale

compound screen in parallel using a wildtype mouse myoblast line, in which the control of differentiation is tightly controlled by the operator. We subsequently assessed the human FKRP iPSC-derived neural cells for increased functional glycosylation of α -dystroglycan with the lead compound. This was the first unbiased large-scale screen (31,954 compounds), in which 4BPPNit was the only compound that could be independently validated using human FKRP-deficient neural cells. Our screen was unbiased hence we did not enrich for compounds that could up-regulate *LARGE1*. As we now have established the first FKRP dystroglycanopathy human iPSC models, future studies combining our human iPSC models and larger compound library screening hold significant promise to identify more active compounds for augmenting functional glycosylation of α -dystroglycan.

2nd Editorial Decision

2 August 2019

Thank you for submitting the revised version of your manuscript. It has now been seen by two of the original referees.

As you can see, both referees find that the study is significantly improved during revision and recommend publication pending satisfactory minor revision. They both raise a significant remaining concern that needs to be addressed. I further discussed this issue with the referees and they both find that the absolute changes in glycosylation levels in response to drug treatment would have to be reported in the figure and in the text and its functional relevance must be discussed.

- Please address the remaining concerns of the referees.
- "Competing interests" should be renamed to "Conflict of interest".
- Figures should be removed from the manuscript and only legends be kept.
- "Materials and Methods" should be moved before "Acknowledgements".
- We noticed that the reference format is currently incorrect. In the reference list, the authors' surnames and initials should be inverted; 'et al' should be used if there are more than ten authors. Please see the author guide for further details:
<https://www.embopress.org/page/journal/14693178/authorguide#referencesformat>
- Our production/data editors have asked you to clarify several points in the figure legends (see attached document). Please incorporate these changes in the attached word document and return it with track changes activated.
- Papers published in EMBO Reports include a 'Synopsis' to further enhance discoverability. Synopses are displayed on the html version of the paper and are freely accessible to all readers. The synopsis includes a short standfirst - written by the handling editor - as well as 2-5 one sentence bullet points that summarise the paper and are provided by the authors. I would therefore ask you to include your suggestions for bullet points.
- Please provide an image for the synopsis. This image should provide a rapid overview of the question addressed in the study but still needs to be kept fairly modest since the image size cannot exceed 550x400 pixels.
- EMBO Press is pleased to support the "minimum reporting standards in the life sciences" initiative (<https://osf.io/preprints/metaarxiv/9sm4x/>). This effort brings together a number of leading journals and reproducibility experts to develop minimum expectations for reporting information about Materials (including data and code), Design, Analysis and Reporting (MDAR) in published papers. We believe broad alignment on these standards will be to the benefit of authors, reviewers, journals and the wider research community and will help drive better practise in publishing reproducible research. We are therefore participating in a community pilot involving a small number of life science journals to test the MDAR checklist. The checklist is intended to help authors, reviewers and editors adopt and implement the minimum reporting framework. Since your manuscript fits the scope of the study, we very much hope that you will be willing to participate in this trial; the MDAR reporting checklist and an MDAR elaboration document providing context for the standards is attached. If you agree to participate, please complete the MDAR reporting checklist and return it to us within 7 days. We would also be very grateful if you could complete this author survey <https://forms.gle/FRx7hpKS8g1QMNPR9>.

Please note that your completed checklist and survey will be shared with the minimum reporting standards working group. However, the working group will not be provided with access to the

manuscript or any other confidential information including author identities, manuscript titles or abstracts. Feedback from this process will be used to consider next steps, which might include revisions to the content of the checklist. Data and materials from this initial trial will be publicly shared in September 2019. Data will only be provided in aggregate form and will not be parsed by individual article or by journal, so as to respect the confidentiality of responses.

Please treat the checklist and elaboration as confidential as public release is planned for September 2019.

Thank you again for giving us to consider your manuscript for EMBO Reports, I look forward to your minor revision.

REFEREE REPORTS

Referee #1:

The authors have addressed the concerns that I raised in the first round of review. While the response to the first question (i.e. changes in other glycosylation components) makes the paper more complete, I am a bit concerned by the answer to the second question (i.e. my concern that from the way that things were presented in figure 6, it was impossible to get an impression of the relevance of the improved glycosylation)

Looking at the full blots (now provided in Appendix Figure S5, it is clear that the glycosylation achieved upon treatment with their compound is very, very far from normal. This puts the presented 2x increase in glycosylation (presented in figure 6) in a quite different context.

I really appreciate the effort and the presentation of the rest of the paper. Yet, I very much dislike the way that figure was and is presented (i.e. fold change to untreated condition). I fear that a reader who doesn't scrutinize the paper will get the impression that there is quite a dramatic change (even though the authors now point towards the different exposure times). Yet, this is not the case. To me this will probably make that the relevance of the effect will be easily overestimated.

Even though this is unorthodox, I would have appreciated to see in Figure 6 some presentation of the relative laminin overlay and/or IHH6 western blot signal in FKRP-knockout and rescued cell lines. The authors state that it is not possible to quantify this together. Yet, they do quantify things in figure 4. Maybe one could just plot the "untreated" conditions in the same way as in figure 4 ?

Referee #2:

It was interesting to see that several of my initial concerns were also raised by reviewer 1. Regarding one of those comments, the description of the effect of 4BPPNt on IHH6 glycosylation, I suggest that the authors will specify the extent of the effect more clearly in the text. So not only saying that it is a modest effect (or significant at other places in the text), but explain what that actually means: and increase from 1 to 2 % of normal levels. Other than that, I am satisfied with the responses of the authors and the clarifications that have been added to the text and the supplementary information.

2nd Revision - authors' response

24 August 2019

Response to referee comments

Referee #1:

The authors have addressed the concerns that I raised in the first round of review. While the response to the first question (i.e. changes in other glycosylation components) makes the paper more

complete, I am a bit concerned by the answer to the second question (i.e. my concern that from the way that things were presented in figure 6, it was impossible to get an impression of the relevance of the improved glycosylation)

Looking at the full blots (now provided in Appendix Figure S5, it is clear that the glycosylation achieved upon treatment with their compound is very, very far from normal. This puts the presented 2x increase in glycosylation (presented in figure 6) in a quite different context.

I really appreciate the effort and the presentation of the rest of the paper. Yet, I very much dislike the way that figure was and is presented (i.e. fold change to untreated condition). I fear that a reader who doesn't scrutinize the paper will get the impression that there is quite a dramatic change (even though the authors now point towards the different exposure times). Yet, this is not the case. To me this will probably make that the relevance of the effect will be easily overestimated.

Even though this is unorthodox, I would have appreciated to see in Figure 6 some presentation of the relative laminin overlay and/or IIH6 western blot signal in FKRP-knockout and rescued cell lines. The authors state that it is not possible to quantify this together. Yet, they do quantify things in figure 4. Maybe one could just plot the "untreated" conditions in the same way as in figure 4?

Referee #2:

It was interesting to see that several of my initial concerns were also raised by reviewer 1. Regarding one of those comments, the description of the effect of 4BPPNIt on IIH6 glycosylation, I suggest that the authors will specify the extent of the effect more clearly in the text. So not only saying that it is a modest effect (or significant at other places in the text), but explain what that actually means: and increase from 1 to 2 % of normal levels.

Other than that, I am satisfied with the responses of the authors and the clarifications that have been added to the text and the supplementary information.

We thank both Referee #1 and #2 for their constructive comments. We have added the immunoblot from the Appendix Fig. S5A to Figure 6A and quantified the relative levels of IIH6 immunoreactivity. It is clear that the effects of 4BPPNIt in FKRP mutant cells (chemical correction) is weak, when compared with the IIH6 immunoreactivity in CRISPR-corrected cells (genetic correction).

Since the signal intensity in the lane of CRISPR-corrected cells is already saturated, we think it is not accurate to specify the percentage of IIH6 immunoreactivity in 4BPPNIt-treated FKRP cells in relation to CRISPR-corrected cells. To make it clearer, we have modified the text to read:

“Compared with untreated control, increased IIH6 reactivity was detected in the 4BPPNIt-treated FKRP-NSCs, but much weaker than that in CRISPR-corrected NSCs (Fig. 6A; Appendix Fig. S5A). Under the same exposure time, the signal intensity of IIH6 reactivity is saturated in corrected-NSCs and its relative level is underestimated (Fig. 6A). Thus, we decided to quantify the effects of 4BPPNIt treatment in FKRP-NSCs and CRISPR-corrected NSCs separately (Fig. 6B-G).”

The functional relevance is reported as below:

“Note that the weak increase of IIH6 reactivity induced by 4BPPNIt (Fig. 6A; Appendix Fig. S5A) was not sufficient to shift the molecular weight of α -dystroglycan core protein in FKRP-NSCs (Appendix Fig. S5B). Together, these data indicate that 4BPPNIt is capable of inducing tissue-specific functional glycosylation of α -dystroglycan in human FKRP-iPSC derived NSCs, although the effect of 4BPPNIt is much weaker than the CRISPR correction of FKRP.”

In the Discussion. We added the following text.

“Although 4BPPNit treatment augmented functional glycosylation of α -dystroglycan (chemical correction), the effect is not sufficient to shift the molecular weight of α -dystroglycan core protein to normal levels as in CRISPR-corrected cells (genetic correction).”

In addition, figure legends have been revised accordingly and comments from the editorial offices have been addressed.

Corresponding Author Name: Yung-Yao Lin

Manuscript Number: EMBOR-2019-47967V1